# Yield-Related QTL Clusters and the Potential Candidate Genes in Two Wheat DH Populations

**DOI:** 10.3390/ijms222111934

**Published:** 2021-11-03

**Authors:** Jingjuan Zhang, Maoyun She, Rongchang Yang, Yanjie Jiang, Yebo Qin, Shengnan Zhai, Sadegh Balotf, Yun Zhao, Masood Anwar, Zaid Alhabbar, Angéla Juhász, Jiansheng Chen, Hang Liu, Qier Liu, Ting Zheng, Fan Yang, Junkang Rong, Kefei Chen, Meiqin Lu, Shahidul Islam, Wujun Ma

**Affiliations:** 1Australian-China Joint Centre for Wheat Improvement, Agricultural Sciences, College of Science, Health, Engineering and Education, Murdoch University, South Street, 90, Murdoch, WA 6150, Australia; J.Zhang@murdoch.edu.au (J.Z.); M.She@murdoch.edu.au (M.S.); R.Yang@murdoch.edu.au (R.Y.); yanjie.j@foxmail.com (Y.J.); qyb.leaf@163.com (Y.Q.); zsn19870322@163.com (S.Z.); Sadegh.Balotf@utas.edu.au (S.B.); zhaoy47249@126.com (Y.Z.); M.Anwar@murdoch.edu.au (M.A.); alhabarzaid@yahoo.com (Z.A.); a.juhasz@ecu.edu.au (A.J.); jshch@sdau.edu.cn (J.C.); Hang.Liu@murdoch.edu.au (H.L.); Qier.Liu@murdoch.edu.au (Q.L.); tingzheng521@hotmail.com (T.Z.); fan.yang@murdoch.edu.au (F.Y.); S.Islam@murdoch.edu.au (S.I.); 2Department of Field Crops, College of Agriculture and Forestry, Mosul University, Mosul 41002, Iraq; 3College of Agriculture and Food, Zhejiang Agriculture and Forestry University, Hangzhou 311300, China; junkangrong@126.com; 4SAGI West, Faculty of Science and Engineering, Curtin University, Bentley, WA 6102, Australia; kefei.chen@curtin.edu.au; 5Australian Grain Technologies, Newell Highway, 12656, Locked Bag 1100, Narrabri, NSW 2390, Australia; Meiqin.Lu@agtbreeding.com.au

**Keywords:** consensus map, correlation, DH populations, grain yield-related traits, QTL, QTL cluster

## Abstract

In the present study, four large-scale field trials using two doubled haploid wheat populations were conducted in different environments for two years. Grain protein content (GPC) and 21 other yield-related traits were investigated. A total of 227 QTL were mapped on 18 chromosomes, which formed 35 QTL clusters. The potential candidate genes underlying the QTL clusters were suggested. Furthermore, adding to the significant correlations between yield and its related traits, correlation variations were clearly shown within the QTL clusters. The QTL clusters with consistently positive correlations were suggested to be directly utilized in wheat breeding, including 1B.2, 2A.2, 2B (4.9–16.5 Mb), 2B.3, 3B (68.9–214.5 Mb), 4A.2, 4B.2, 4D, 5A.1, 5A.2, 5B.1, and 5D. The QTL clusters with negative alignments between traits may also have potential value for yield or GPC improvement in specific environments, including 1A.1, 2B.1, 1B.3, 5A.3, 5B.2 (612.1–613.6 Mb), 7A.1, 7A.2, 7B.1, and 7B.2. One GPC QTL (5B.2: 671.3–672.9 Mb) contributed by cultivar Spitfire was positively associated with nitrogen use efficiency or grain protein yield and is highly recommended for breeding use. Another GPC QTL without negatively pleiotropic effects on 2A (50.0–56.3 Mb), 2D, 4D, and 6B is suggested for quality wheat breeding.

## 1. Introduction

Wheat (*Triticum aestivum*) is grown on 17% of the world’s cultivated land, which is equivalent to 220 million hectares. There is 750 million tons of annual wheat production (www.fao.org/faostat, 2014; accessed on 13 Febuary 2020), supplying food for 40% of the human population [1]. In Australia, around 22 million tons of wheat is produced annually, and 70% is exported, accounting for 11% of the world trade (Australian Agricultural Trade, www.ruralbank.com.au, accessed on 13 Febuary 2020). An increase of grain yield by 60% by 2050 will be required due to the world’s growing population [2]. Over the last few decades, breeding high yield varieties has been one of the most important approaches for yield increase, as most agronomic traits show high heritability and are controlled by genes. For tracking those agronomic traits, quantitative trait loci (QTL) mapping and genome wide association studies (GWAS) have become major strategies to identify trait related molecular markers [3,4,5,6,7].

Genetic linkage map construction is essential for QTL mapping. Recently, high-throughput SNP markers have been widely used in wheat genetic map construction due to the high marker density and low cost [8,9,10]. Consensus maps with higher marker density and greater genome coverage have been developed through integrating genetic markers of multiple populations [11,12]. QTL mapping with multiple populations would provide more robust results based on the enriched genetic background and high level of polymorphisms [13].

The frequently targeted agronomic traits for QTL analysis are usually grain yield-related phenotypes, such as anthesis time, plant height (PH), grain weight per tiller (GW), thousand grain weight (TGW), grain number per spike (GN), spikelet number per spike, biomass, spike length, and grain protein content (GPC), which are targeted by wheat breeders [14].

Flowering time is crucial for high yield and environmental adaptation, as it is associated with the time of spikelet differentiation and floret development, and grain filling time. Breeders tend to breed wheat varieties with early flowering and maturity to avoid terminal drought and heat stresses; however, those varieties more likely encounter frost damage [8,15]. Therefore, proper flowering genes need to be carefully selected. Flowering-related genes mainly include vernalization (*VRN*), photoperiod (*Ppd*), and earliness per se (*Eps*). The *VRN* genes include *VRN1*, *VRN2*, and *VRN3* in groups 5 and 7 [16,17,18]. *VRN4* (*VRN-D4)* is a homologous gene of *Vrn-A1* which was identified in the short arm and close to the centromeric region of chromosome 5D [19]. Another flowering gene *TaVrt2* in group 7 was identified that regulates the floral transition [20,21].

Within the *PPD1* genes on group 2 [22], the homologous genes *Ppd-A1* and *Ppd-B1* showed lesser effects on flowering during short days than the *Ppd-D1* [23]. However, the high copy number of *Ppd-B1* strongly influenced the flowering time [8,24]. *Ppd-A1* tends to increase TGW and yield, whereas *Ppd-B1* seems to be associated with a high kernel number through increasing the spikelet number [25].

In comparison with genes of vernalization and photoperiod, the influence of *earliness per se* (*eps*) genes showed lower-level effects on early flowering but high levels of epistatic interaction with *Ppd1* [26,27,28]. *Eps* loci are associated with spikelet number, thereby affecting wheat yield [26]. Numerous eps and the related flowering-time QTL in wheat have been mapped to chromosomes 1DL, 2B, 3A, 4A, 4B, and 6B [29,30,31]. In wheat, a homolog to Arabidopsis’s *Early flowering 3* (*ELF3*) gene on chromosome group 1 was identified as a candidate gene of Eps-Am1. TaELF3-1DL tends to be the major isoform of gene TaELF3 [32,33].

Besides the well-known anthesis-related genes above, a set of *T*. *aestivum* gigantea (*TaGI*) encoding genes are located on group 3, which interact with flavin-binding, kelch repeat, and F-Box 1 (FKF1) domains to form a complex regulating photoperiod-dependent flowering by regulating constant (CO) expression [34,35]. A *SOC1* (*suppressor of overexpression of CO 1*)-like gene on chromosome 4DL, *WSOC1*, also influences flowering time in wheat [36]. There are three other short-day flowering-time genes, including *flowering locus T3* (*TaFT3-B1*), *WUSCHEL-like* (*TaWUSCHELL-B1*), and *Target of Eat1* (*TaTOE1-B1*) on 1B [37]. A set of heading-date genes (*TaHD1*) was identified on group 6, interacted with vernalization genes under long-day conditions [38,39].

Plant height also heavily affects the grain yield. About 25 height reducing genes (*Rht1*-*Rht25*) have been identified in wheat; and *Rht1*, *Rht2*, *Rht8*, and *Rht12* have been used in wheat breeding even though most of them need further studies for their benefits on agronomic traits [40]. *Rht_NM9* and *Rht7* are on 2AS [41,42]; *Rht4*, *Rht8*, and *Rht5* on 2BL, 2DS, and 3BS, respectively [43]; and *Rht1* (*Rht-B1b*) on 4BS [44,45]. *Rht3* (*Rht-B1c*), *Rht1S* (*Rht-B1d*), *Rht11* (*Rht-B1e*), *Rht T.aeth* (*Rht-B1f*), *Rht17* (*Rht-B1p*), and *Rht_107* were found to be allelic to *Rht1* [46,47,48,49]; *Rht2* (*Rht-D1b*) on 4DS; *Rht10* (*Rht-D1c*) and *Rht Aibian 1a* (*Rht-D1d*) were allelic to *Rht2* [45,50,51]. *Rht12* and *Rht9* were on 5AL [43]; *Rht23* on 5DL [52]; *Rht18* on 6AL. *Rht14*, *Rht16*, and *Rht24* were found to be allelic to *Rht18* and *Rht15* on 6AS [47,53]. *Rht22* and *Rht13* were on 7AS and 7BS, respectively [43,54]. The locations of *Rht6*, *Rht15*, *Rht19*, and *Rht20* have not been identified [40].

The QTL of other agronomic traits, such as TGW, GW, GN, spikelet number per spike, biomass, spike length, GPC, and seed parameters have been identified in almost all 21 chromosomes in wheat [4,5,6,7,11,55]. A few yield-related genes have been cloned. For example, *TaGW2* gene on 6A was cloned and found to have a 6.6% contribution to grain weight [56,57]. Grain size (GS)-related genes were identified, such as *TaGS5-3A* [58], *TaTGW6-4AL* [59], and *TaTGW-7A* [60]. A gene for grain length (*TaGl3-5A*) was cloned, and its effect on grain length was identified [61]. There are two paralogous groups of *GPC1* and *GPC2* located on chromosome groups 6 and 2 in wheat, respectively. *GPC1* showed a key regulator of nutrient remobilization during the early stages of senescence [62]. The wild-type allele with high GPC was identified in a wild emmer wheat (*Triticum turgidum* subsp. *dicoccum*) in Israel [63]. The sequence of *GPC1-B1* revealed that it encoded a NAC (NAM, ATAF, and CUC)-domain transcription factor, and a non-functional copy is present in the most modern wheat varieties [64]. It was also reported that natural variants of *GPC1-A1* have the same role as *GPC1-B1* [65,66]. Recently, GPC QTL was further validated on 2A and 2B, and *Fd-GOGAT_2A* and *GS2-B2* (glutamine synthetase) were suggested as candidates for those QTL, respectively [67].

The current study aimed to identify the genetic basis of yield-related traits in two double haploid (DH) populations, Bethlehem x Westonia (BW) and Spitfire x Bethlehem (SpB), which share one common parental line. Twenty two phenotypes were investigated, including grain yield, GN, TGW, GPC, protein yield (PY), nitrogen use efficiency (NUE), biomass per m^2^ (BioSq), biomass per tiller (BioTiller), grain weight per m^2^ (GW), grain weight per tiller (GWTiller), spikelet number per spike (Spikelet), days to anthesis (Anthesis), plant height (Height), peduncle length (PedLength), stem length (StLength), peduncle ratio (PedRatio), spike length (SpLength), keycard dimeter (Keycard), seed area (SdArea), seed length (SdLength), seed thickness (SdThick), and seed width (SdWidth). A 12k targeted genotyping-by-sequencing (tGBS) assay designed on a subset of the 90k SNPs in the Infinium beadchip was used to construct genetic map for the QTL analysis of 22 yield-related traits from four large-scale field trials. Our endeavors allowed us to identify robust QTL clusters with repetitive QTL in two populations and the linkages of yield-related traits within the QTL clusters, to search for yield-related candidate genes, and to highlight new relationships between traits and genetic loci for wheat breeding.

## 2. Results

### 2.1. Phenotypic Variations

Among the parental lines, the common parent Bethlehem showed lower grain yield, and high TGW and GPC (Figure 1 and Appendix A). Due to the grain yield being low, it showed the lowest protein yield (PY) and NUE. On the other hand, Westonia presented the highest grain yield and relatively low TGW and GPC. Spitfire showed the highest GN, PY, and NUE. The highest GN of Spitfire was probably due to the highest spikelet number per spike (*p* < 0.05) which may stem from the late anthesis, as Bethlehem flowered one or two days earlier than Spitfire and Westonia. Westonia was slightly earlier than Spitfire. The higher GN also resulted in the highest grain weight per tiller in Spitfire. Westonia showed the highest biomass and grain weight in one m^2^ area, which indicate the tiller number per m^2^ was relatively high, as the GW per tiller was not the highest. Westonia also showed the lowest plant height with the lowest values of stem length (*p* < 0.05), peduncle length, and spike length, whereas its peduncle ratio was the highest. Spitfire showed the lowest keycard diameter (Figure 1 and Appendix A). For the seed parameters, the seed length of Spitfire was the longest (*p* < 0.05) and the seed width was relatively lower. The seeds of Bethlehem were thicker than Westonia and Spitfire, but no significant difference was obvious in seed area (Appendix A).

The phenotype mean values of replicates in each environment were used for violin plots of each DH population. The phenotype distributions of two DH populations in each environment are shown in Figure 2 and Appendix A. The phenotypic values of both populations showed similar patterns in different environments, except the phenotypes of the peduncle ratio, spike length, and spikelet number per spike (Appendix A). On average, the grain yield in Narrabri was the highest, followed by Williams, Beverley, and Muresk for both populations (Figure 2). The TGW in Beverley was the lowest. The lowest GPC was in Williams. The protein yield and NUE showed the same patterns (Figure 2 and Appendix A). Both populations flowered late in Muresk and Williams in 2018. Flowering was about 20 days early in Beverley in 2020 (Figure 2 and Appendix A).

The detailed grain yield components were taken from the field trials in Muresk and Williams in 2018 (Appendix A). Compared with Muresk, the average value of biomass, biomass per tiller, grain weight per m^2^, grain weight per tiller, GN, and spikelet number per spike were much higher in Williams (Appendix A). The values of the plant height, stem length, peduncle length, and keycard diameter were also higher in Williams (Appendix A). The spikelet number per spike in Narrabri was similar to that in Williams. From the view of grain yield and yield components, those results showed the fields of Williams and Narrabri were favorable for plant growth. The peduncle ratios were similar between populations and two environments (Muresk and Williams). The spike length of SpB was higher than that of the BW population. For the seed parameters, the seed area in Beverley was slightly less, which may have been due to the lower levels of seed thickness and seed width, although the seed length in Beverley was slightly higher (Appendix A).

### 2.2. Phenotypic Correlations in Each Environment

The pairwise correlations between those 22 phenotypes plus grain weight per plant are shown in Figure 3 and Appendix A. Significantly positive correlations appeared between grain yield and yield components, such as, GY, PY, NUE, biomass, GW, spikelet number per spike, and GN, in both populations, in two or more environments. Significantly positive correlations between GY and plant height were found in the SpB at Williams and Muresk, whereas that correlation was found at Williams only for BW. The GY also showed significantly positive correlations with the days to anthesis in the SpB population in three locations, Williams, Muresk, and Narrabri (negative to Zadok stages). Expectedly, the TGW showed significantly positive correlations with seed parameters, such as seed area, seed length, seed thickness, and seed width in almost all environments in both populations; and with the plant height, stem length, and peduncle length at Muresk and Williams. The GPC expectedly had significantly negative correlations with GY in both populations and all environments; and with biomass, grain weight, GN, and days to anthesis at Muresk and Williams. GPC was positively correlated with seed length in both populations, and TGW, seed area, seed thickness, and seed width in BW only at Williams. At Muresk and Williams, the GN was also negatively associated with the TGW and seed related parameters, but positively associated with GY and days to anthesis in both populations. The peduncle ratio was negatively associated with spike length, and stem length, PY, GY, and grain yield components. The new parameter (Keycard) was positively correlated with spikelet number per spike in both populations at Muresk and Narrabri, and with GY at Williams and Muresk.

### 2.3. Consensus Map Construction

For QTL cluster analysis in two DH populations, a consensus map was generated. For BW, 2387 markers were formed (736 bin markers) and allocated into 28 linkage groups. Two linkage groups each were on 1D, 2D, 6D, and 7D, and four were on 7A. The total genome coverage was 3552.67 cM with an average bin marker interval of 4.83 cM. The minimum average bin marker interval was on 5B (3.2 cM), and the maximum was on 5D (16.76 cM) (Appendix A).

For SpB, 23 linkage groups were formed by 2572 markers with two linkage groups each on 2D and 4D. The total genome coverage was 3220 cM with an average bin marker interval of 3.94 cM. The minimum and maximum of average bin marker intervals were on 1B (2.40 cM) and 6D (18.45 cM) (Appendix A).

The consensus map (BW–SpB) was constructed by calculating the consensus marker orders of linkage groups according to the individual maps. A total number of 3583 markers were grouped into 22 linkage groups. Two linkage groups were on 2D. The total coverage of the consensus map (BW–SpB) was 3539.29 cM, with an average bin marker interval of 3.74 cM. The total map length for each chromosome ranged from 87.23 cM (1D) to 252.97 cM (5A), and the minimum average bin marker interval was on 3A and 5B (2.50cM). The maximum was on 3D (12.81 cM). The results showed the SNP markers were enriched and the average bin marker interval was reduced in the BW–SpB consensus map (Appendix A). The circus graph of collinearity among the BW–SpB consensus map, the physical map (IWGSC RefSeq v1.0), and the consensus map of Wang et al. (2014) [68] is presented in Figure 4. The collinearity among the individual genetic maps (BW or SpB), the BW–SpB consensus map, and the physical map are shown in Appendix A.

### 2.4. Robust QTL Identification in Two Populations

Due to the high correlations between different traits, and very similar QTL results, the QTL of some traits have been combined—for example, protein yield (PY) and NUE, and plant height and stem height. Grain weight per m^2^ and grain weight per tiller were combined as grain weight; and biomass per m^2^ and biomass per tiller were combined as biomass. The QTL on the same or close regions (Appendix A) were also combined, even though they were detected from replicates and different environments. In Appendix A, the QTL of 18 traits were summarized, and a total of 227 QTL are presented.

#### 2.4.1. QTL for Anthesis

The days to anthesis parameter determines the lengths of vegetative and productive growth stages, and was significantly associated with grain yield and GPC. During the QTL grouping, the Zadok stage was incorporated with anthesis. In four environments, 10 anthesis QTL were emerged, and seven of them were detected in multiple environments or replicates which were on 3A, 3B, 4B, 5A, 5D, 7B, and 7D (Appendix A). The QTL on 2A, 4B’s short arm, and 5B were only found in one replicate. The LOD values of QTL on 3A, 4B, 5A, and 7B were above four and showed high phenotypic contributions. Out of the 10 QTL, five QTL were contributed by Bethlehem, which were on 2A, 3A, 3B, 4B’s short arm, and 7B, whereas the other five QTL were contributed by either Westonia or Spitfire. In particular, the QTL on 5A and 5D overlapped with gene *Vrn1_5A* and *Vrn1_5D*, respectively. Tthe QTL on 5B was close to the region of *Vrn1_5B* (Appendix A).

#### 2.4.2. QTL for Grain Yield and Grain Weight

There were 11 QTL detected for grain yield on eight chromosomes, including 1A, 2B, 4A, 4B, 5B, 7A, 7B, and 7D. Five QTL on 2B, 4A, 7A, 7B, and 7D were repeatedly detected in multiple environments or replicates. The high LOD value (>4) of grain yield QTL appeared on 2B, 7A, and 7D and reached 20% or more phenotypic variations. The former two were contributed by Spitfire and the latter one by Westonia.

There were 18 GW QTL detected on 12 chromosomes, including 1B, 1D, 2A, 2B, 3B, 4A, 5A, 5B, 5D, 7A, 7B, and 7D. Six QTL on 2B, 5D, 7A, 7B, and 7D were repetitively detected in multiple environments or replicates, with one contributed by Bethlehem, two by Spitfire and three by Westonia. The QTL on 7B and 7D were appeared in similar region in the physical map in both populations (Appendix A). Eight QTL with high LOD value (>4) were present on 1B, 2B, 3B, 7A, 7B, and 7D with three each contributed by Bethlehem and Spitfire, and two by Westonia.

#### 2.4.3. QTL for GPC, NUE and Protein Yield

A total of 12 GPC QTL were detected on nine chromosomes, including 2A, 2B, 2D, 4D, 5B, 6B, 7A, 7B, and 7D, though only one on 7A was present in multiple environments. There were six QTL on 2A, 2B, 5B, and 7A with high LOD value (>4) and four of them were contributed by Bethlehem, and one each by Westonia and Spitfire.

Due to the calculation of NUE and protein yield (PY), the QTL of these two parameters were same. There were 12 QTL that appeared in a single environment, which was in agreement with the relatively low heritability of NUE/PY (BW: 0.76, SpB: 0.66). Those QTL were located on eight chromosomes, including 1A, 1B, 1D, 2B, 3B, 5A, 5B, and 7B. The QTL on 1A were detected in the same region in both populations and contributed by Spitfire and Westonia. The other five QTL were contributed by Bethlehem and four by spitfire and two by Westonia. The large QTL (LOD > 4) appeared on 1A, 2B, 3B, 5A, and 7B, and accounted for 14–31% of the phenotypic variation (Appendix A).

#### 2.4.4. QTL for Biomass

Only six QTL of biomass were detected on 1D, 2B, 3B, 3D, 7A, and 7D in the SpB population, and all appeared in a single environment (Appendix A), which agrees with the low heritability (BW: 0.35, SpB: 0.6). The QTL on 2B were present in replicates at Williams. Most QTL were contributed by Spitfire except for the QTL on 3B by Bethlehem. The QTL with high LOD values of 6.1 and 8.6 were on 2B and 3D, respectively.

#### 2.4.5. QTL for GN and Spikelet Number per Spike

Eleven GN QTL were detected on seven chromosomes, including 1B, 2B, 3A, 4A, 5A, 7A, and 7B—three contributed by Bethlehem, four by Spitfire, and four by Westonia. Only two QTL on 2B were detected repetitively in Williams. The QTL with the high LOD value (>4) was on 7A and contributed by Spitfire (Appendix A).

There were 11 QTL of spikelet number per spike detected on 10 chromosomes, including 1B, 2D, 3A, 3B, 3D, 5A, 5B, 5D, 6A, and 7B; and six of those were contributed by Bethlehem, three by spitfire, and two by Westonia (Appendix A). The QTL on 2D in both populations were on the neighboring markers on the consensus map, and were considered as the same QTL (Appendix A). QTL on 1B, 2B, 2D, 5B, and 5D were detected in multiple environments or replicates. Six QTL with high LOD values (>4) were on 1B, 2B, 3D, 5B, 5D, and 7B—three QTL are attributed to Spitfire, two to Bethlehem, and one to Westonia.

#### 2.4.6. QTL for TGW and Seed Parameters

For TGW, 11 QTL were on 1A, 1B, 4A, 4B, 4D, 5A, 5B, 7A, and 7B. The most QTL were contributed by Bethlehem, and three by Westonia on 4D, 5B, and 7B. Except 4D, 5A, 7A, and 7B, all other QTL were detected in multiple environments or replicates. Particularly, the QTL on 1B were detected in both populations and considered as one QTL, as these two QTL were close to each other on the consensus map and physical map (Appendix A). The QTL with LOD values exceeding four were on 1A, 1B, 4A, 4B, 5A, and 7B.

There were 22 QTL of seed area detected on 13 chromosomes, including 1A, 1B, 2A, 2B, 2D, 3A, 4A, 4B, 5A, 5B, 6A, 7A, and 7B—13 contributed by Bethlehem, 3 by Spitfire, and 6 by Westonia. Out of those, five QTL on 1A, 2A, 5A, 6A, and 7B appeared in multiple environments or replicates. Nine QTL on 1A, 1B, 2A, 4A, 4B, 5A, 7A, and 7B had high LOD values (>4) (Appendix A).

For seed length, 14 QTL were detected on 10 chromosomes, including 1B, 2A, 2B, 2D, 3A, 3D, 4A, 5A, 7A, and 7B—nine contributed by Bethlehem, two by Spitfire, and three by Westonia. Interestingly, most QTL were detected repetitively in multiple environments or replicates, except on the long arm of 2A, 2D, the short arms of 2B and 5A, 7A, and 7B (Appendix A). The results matched with the high heritability of seed length in both BW (0.95) and SpB (0.97) populations.

For seed thickness, 14 QTL were detected on 11 chromosomes, including 1A, 1B, 3B, 4A, 4B, 4D, 5A, 5B, 6A, 6B, and 7A—11 contributed by Bethlehem, one by spitfire, and two by Westonia. Five of those on 1A, 4A, 4B, 5B, and 7A were detected in multiple environments and replicates. Six QTL on 1A, 1B, 4A, 4B, 6B, and 7A showed high LOD values over 4 (Appendix A).

For seed width, 23 QTL were detected on 15 chromosomes, including 1A, 1B, 1D, 2A, 2B, 3A, 3B, 4A, 4B, 5A, 5B, 5D, 6A, 7A, and 7B—13 QTL contributed by Bethlehem, three by Spitfire, and seven by Westonia. The QTL on 5D were on the *Vrn1_5D* region in both populations and were considered as one QTL (Appendix A). Six QTL on 1A, 1B, 4B, 5A, 5D, and 7B were present in multiple environments or replicates. There were 12 QTL on 1A, 1B, 3B, 4A, 4B, 5A, 5B, 5D, 7A, and 7B which showed high LOD values (>4) (Appendix A).

#### 2.4.7. QTL for Plant Height and Stem Length

The QTL of plant height and stem length were combined in analysis. In total, 18 QTL were detected on 13 chromosomes, including 1B, 1D, 2A, 2B, 2D, 3A, 3B, 4A, 5A, 6A, 6B, 7A, and 7B—10 contributed by Bethlehem, seven by Spitfire and one by Westonia (Appendix A). The QTL on 1B’s short arm in both populations were 9 cM apart in the consensus map (Appendix A) and were considered as one QTL. Seven QTL on 2B, 3A, 3B, 4A, 5A, and 7B were appeared in multiple environments or replicates. Most height QTL presented high LOD values (>4) except QTL on 1B, 4A, 6B, and 7A (Appendix A).

#### 2.4.8. QTL for Peduncle Length and Peduncle Ratio

There were seven QTL of peduncle length detected on six chromosomes, including 1B, 2B, 4A, 4B, 5A, and 7D—three contributed by Bethlehem, three by Spitfire, and one by Westonia. Three QTL on 2B, 4A, and 7D appeared in multiple environments or replicates. Four QTL on 2B, 4B, 5A, and 7D showed high LOD values (>4) (Appendix A).

Seven QTL of peduncle ratio were detected on 1D, 2A, 3A, 3B, 4B, and 5D—three contributed by Bethlehem, two by Spitfire, and two by Westonia. Interestingly, none of those QTL were detected in multiple environments, which explains the relatively low heritability in BW (0.61) and SpB (0.54) populations. Three QTL showed high LOD values over 4, and the highest LOD value was 9 on 4B, attributed to Westonia.

#### 2.4.9. QTL for Spike Length and Keycard Diameter

Five QTL of spike length were detected on four chromosomes, including 1A, 2B, 5A, and 6D—four contributed by Bethlehem and one by Spitfire. The two QTL on 5A were detected repeatedly in replicates at Williams, and one of those showed a high LOD value of 4.6 (Appendix A).

Eight QTL of keycard diameter appeared on six chromosomes, including 1B, 2A, 2B, 3D, 5A, and 5B. Three were contributed by Bethlehem, four by Spitfire, and one by Westonia. Only one QTL on 3D was detected in multiple environments. The QTL on the short arm of 2B and 3D showed high LOD values (>4), and both were attributed to Spitfire (Appendix A).

### 2.5. QTL Clusters and Related Potential Candidate Genes

In the combination of QTL detected in BW and SpB populations, 35 QTL clusters were identified on 18 chromosomes, except 3D, 6B, and 6D (Appendix A). The cluster locations in the BW–SpB consensus map and physical map, and the first and last genes within the cluster, followed after each cluster. In each QTL cluster region, the following aspect related genes were selected (Figure 5, Appendix A) and predicted as the potential candidate genes contributing to QTL: anthesis (flowering, *WUSCHEL-like* homeobox), membrane transporters (ABC transporter ATP-binding protein ARB1, transmembrane proteins), hormone-responsive factors (including auxin, abscisic acid, ethylene, and gibberellin), domains related to photoperiod regulators (F-box, CONSTANS), tissue development related transcription factors (GATA, MYB, MADS-box, WRKY), photosystem related proteins (photosystem I or II center proteins, chlorophyll a/b binding proteins), transporters (sugar transporters, sucrose transporters, transporters of nitrate, phosphate, potassium, sulfate, protein transporter related families), carbohydrate synthetic pathways (fucosyltransferase, glycosyltransferase, sucrose phosphate synthase, sucrose synthase), NAC domain proteins, growth regulators, root meristem regulator, seed maturity, and senescence regulators.

Two QTL clusters were identified on 1A. The cluster (1A.1) was located between markers 1AM74868 and 1AM55442 (70.1–79.5 cM, 458.4–494.6 Mb) that included 459 genes (Appendix A). The QTL of TGW, seed area, seed width, and seed thickness were contributed by Bethlehem. The NUE (PY) was contributed by Westonia. The grain yield and NUE (PY) QTL were located on the distal region of the long arm on 1A and formed another cluster (1A.2) between markers 1AM56272 and 1AM10986 (544.1–591.1 Mb), including 920 genes. The QTL were contributed by either Westonia or Spitfire.

Three QTL clusters formed on 1B (Figure 6 and Appendix A). Among them, one (1B.1) was on the short arm between 1BM12602 and 1BM7923 (16.9–36.8 cM, 10.5–39.5 Mb) containing 377 genes, and the QTL included GW, Keycard, and Spikelet. The second cluster (1B.2) was between 1BM79457 and 1BM77253 (65.9–77.7 cM, 456.3–562.0 Mb) containing 888 genes, including the QTL of plant height, TGW, seed area, seed length, seed width, and seed thickness. The third cluster (1B.3), including TGW, seed area, seed width, seed length, plant height, peduncle length, NUE/PY, and GN, was between 1BM1403 and 1BM78633 (116.1–175.3 cM, 630.8–681.3 Mb) containing 845 genes. Interestingly, except the QTL of GN on 1B.3 contributed by Westonia, all other phenotypes were attributed to Bethlehem in two populations.

On 1D, two QTL clusters were formed with one (1D.1) between 1DM61318 and 1DM75370 (20.8–44.3 cM, 7.0-32.5 Mb), including 436 genes and another (1D.2) between 1DM27449 and 1DM77946 (54.0–70.3 cM, 308.5–411.2 Mb), including 1191 genes (Appendix A). The QTL of biomass, plant height, and NUE/PY on 1D.1 were contributed by Spitfire while the seed width and peduncle ratio on 1D.2 by Bethlehem (SpB) and Westonia, respectively.

Two QTL clusters were formed on 2A. The QTL of plant height and seed length formed one cluster (2A.1) between 2AM1628 and 2AM7354 (3.3–34.4 cM, 5.4–32.1 Mb) and were contributed by Spitfire and Bethlehem in SpB and BW populations, respectively (Appendix A). There were 671 genes in this cluster. The second cluster (2A.2) was on the long arm between 2AM11090 and 2AM26001 (111.5–163.0 cM, 711.6–758.0 Mb) containing 1022 genes. The QTL of GW and Keycard were contributed by Westonia, anthesis, and peduncle ratio by Bethlehem in the BW population; seed area and seed length were contributed by Spitfire. Out of those two QTL clusters on 2A, two QTL of seed area and GPC were closely linked to the top region of 2A without the negatively pleiotropic effect in the BW population, and both QTL were contributed by Westonia.

Three QTL clusters were identified on 2B (Figure 7 and Appendix A). The first cluster (2B.1) was located between markers of 2BM23606 and 2BM48834 on the short arm of 2B (52.2–80.0 cM, 44.8–106.0 Mb), including 691 genes. The QTL of GY, GW, GN, Spikelet, NUE/PY, biomass, plant height, and Keycard were contributed by Spitfire, whereas GPC, seed area, and seed length were contributed by Bethlehem. The second cluster (2B.2) was between markers 2BM75364 and 2BM10875 (82.8–89.4 cM, 164.1–381.8 Mb) containing 1048 genes, and the QTL traits of peduncle length, NUE/PY, and seed area, contributed by Spitfire, Westonia, and Bethlehem (BW), respectively. The third cluster (2B.3) was between markers 2BM80969 and 2BM69372 on the long arm of 2B (94.1–133.9 cM, 465.1–721.9 Mb), including 2445 genes. The QTL of GY, GN, and GW were contributed by Westonia; and the plant height, spike length, peduncle length, and seed length were contributed by Spitfire. Seed width and Keycard were contributed by Bethlehem.

There were two QTL clusters detected on 2D (Appendix A). One cluster (2D.1) was between markers 2DM77420 and 2DM53594 (0–23.2 cM, 320.5–523.2 Mb) comprising 1826 genes. The Spikelet QTL were attributed to Bethlehem in both populations, and the seed length to Westonia. Another cluster (2D.2) was detected near the end of 2D chromosome between markers 2DM26538 and 2DM41532 (57.6–80.7 cM, 608.2–777.0 Mb) containing 967 genes. The GPC QTL was contributed by Bethlehem (BW) and seed area by Spitfire.

Two QTL clusters were detected on 3A (Figure 8 and Appendix A). One cluster (3A.1) was between markers 3AM80529 and 3AM56763 (45.6–79.3 cM, 33.2–507.3 Mb), including 1865 genes. The phenotypes of anthesis, GN, Spikelet, and plant height were contributed by Bethlehem; and the peduncle ratio and seed width by Spitfire and Westonia, respectively. Another cluster (3A.2) was between markers 3AM41929 and 3AM366 (175.8–181.7 cM, 714.4–717.9 Mb) comprising 59 genes, where the QTL of seed area and seed length were located, and both phenotypes were attributed to Bethlehem in the BW population.

Two QTL clusters were detected on 3B, one for each population (Appendix A). One QTL (3B.1) cluster was identified between 3BM71944 and 3BM1239 (102.2–135.5 cM, 626.1–736.7 Mb), including 1142 genes in the BW population. The QTL of GW and Spikelet were contributed by Bethlehem, and seed width by Westonia. Another cluster (3B.2) was between 3BM76508 and 3BM28159 (154.5–203.0 cM, 760.7–809.5 Mb) containing 482 genes at the end of 3B in the SpB population, including the QTL of GW and NUE/PY contributed by Bethlehem.

A number of QTL were detected on 3D in both populations, such as biomass, Spikelet, Keycard, and seed length (Appendix A). There was probably a lack of SNP markers on 3D. The QTL cluster was not able to be identified.

On 4A, two QTL clusters were formed (Appendix A). One cluster (4A.1) was between 4AM10748 and 4AM52625 (0–31.9 cM, 3.0–16.7 Mb) on the distal region of 4A’s short arm, including 236 genes. The QTL included GY, GW, GN, seed area, and seed thickness. The second cluster (4A.2) was between markers 4AM9799 and 4AM77998 (83.9–143.4 cM, 601.2–674.9 Mb) comprising 1101 genes in the BW population. The phenotypes included plant height, peduncle length, TGW, seed area, seed length, and seed width. Similarly to the 1B chromosome, most of those QTL on 4A were contributed by Bethlehem, and only seed width by Westonia.

The QTL cluster of grain yield and anthesis contributed by Bethlehem were on the homologous region of 4B between markers 4BM80051 and 4BM80662 (1.4–15.1 cM, 1.3–52.1 Mb), including 697 genes in two populations (Appendix A). There was another QTL cluster between 4BM50833 and 4BM51435 (72.5–114.8 cM, 574.9–665.1 Mb) on 4B containing 1158 genes in the BW population. The QTL of anthesis, peduncle length, and peduncle ratio (1) were attributed to Westonia; and the TGW, seed area, seed thickness, seed width, and another peduncle ratio (2) to Bethlehem.

One QTL cluster was detected between 4DM80718 and 4DM76464 (56.5–61.51 cM, 30.0–150.0 Mb) comprising 1216 genes (Appendix A). The QTL of GPC and TGW were contributed by Bethlehem and Westonia in SpB and BW populations, respectively. No negatively pleiotropic effect was aligned with the GPC QTL.

Three QTL clusters were formed on 5A (Appendix A). One cluster (5A.1) was between markers 5AM43473 and 5AM43262 (52.8–66.1 cM, 441.0–467.4 Mb), including 313 genes. The QTL of Keycard and NUE/PY were contributed by Spitfire, but the plant height, Spikelet, and seed width by Bethlehem. The second cluster (5A.2) was between markers of 5AM27024 and 5AM20138, including 478 genes in the SpB population (99.4–124.3 cM, 515.9–549.4 Mb) containing the QTL of GW, NUE/PY, plant height, peduncle length, and spike length, all attributed to Bethlehem. The third cluster (5A.3) was on the end of 5A’s long arm between markers of 5AM72283 and 5AM76666 (144.0–207.0 Mb, 579.1–653.9 Mb) comprising 1162 genes in the SpB population. The QTL of anthesis and GN were contributed by Spitfire harboring winter-type *Vrn1-5Ab*; and the Keycard, TGW, seed area, seed length, seed thickness, seed width, and spike length by Bethlehem holding spring-type *Vrn1-5Aa*.

Two QTL clusters were identified on 5B (Figure 9 and Appendix A). One cluster (5B.1) was between 5BM78354 and 5BM11567 (55.1–95.2 cM, 368.5–533.0 Mb) containing 1800 genes. The QTL of Spikelet and Keycard were attributed to Spitfire; and the seed area, seed width, and thickness by Westonia. Another cluster (5B.2) was between 5BM46778 and 5BM39285 (156.0-183.7, 613.6–701.5 Mb) including 1302 genes. The QTL of anthesis, NUE/PY, and one GPC were contributed by Spitfire; and the TGW, seed area, seed thickness and seed width by Westonia. The QTL of GW and another GPC were attributed to Bethlehem in BW and SpB populations, respectively. Interestingly, one GPC QTL was parallel with the QTL of days to anthesis, and both were contributed by Spitfire. This result is against the normal findings that the high GPC level is concomitant with the early anthesis.

One QTL cluster was formed on 5D between 5DM62829 and 5DM34532 (9.2–136.5 cM, 338.2–467.2 Mb), including 2081 genes (Appendix A). The QTL of GW and anthesis were contributed by Westonia harboring winter-type of *Vrn1-5Db*; and the seed width by Bethlehem possessing spring-type *Vrn1-5Da* in both populations. *Vrn1-5Da* (TraesCS5D01G401500; 5D: 467.1 Mb) was segregating in the BW population, and those QTL were overlapped with the gene location.

One QTL cluster was formed between 6AM33596 and 6AM7466 (0–9.2 cM, and 1.0–13.8 Mb) on 6A’s short arm containing 286 genes in the SpB population (Appendix A). The QTL of seed area, seed width, and plant height were contributed by Spitfire.

No QTL cluster was identified on 6B and 6D in both populations. The QTL of GPC and seed thickness were positioned on 6B in the BW population; and the QTL of plant height and stem length were on the distal region of the long arm of 6B in the SpB population (Appendix A). No negatively pleiotropic effect was aligned with the GPC QTL. The QTL of spike length was detected on 6D in the SpB population only (Appendix A).

There were two QTL clusters on 7A (Appendix A). One (7A.1) was between 7AM8783 and 7AM81002 (41.5–101.6 cM, 48.9–138.8 Mb), including 1177 genes. The QTL of TGW, seed width, and seed thickness were contributed by Bethlehem (BW) and seed area by Westonia; the GY, GW, GN and biomass by Spitfire; and GPC by Bethlehem in SpB. Another cluster (7A.2) was between 7AM70292 and 7A.2M80464 (178.1–217.2 cM, 697.0–730.3 Mb), comprising 551 genes. The QTL of GY, GW, and seed width were contributed by Spitfire and GPC by Bethlehem in the SpB population, and GN by Westonia.

Two QTL clusters were formed on 7B (Appendix A). One cluster (7B.1) was between 7BM23608 and 7BM76049 (1.8–19.5 cM, 1.8–16.8 Mb) containing 176 genes. The QTL of GY and GW were contributed by Bethlehem; and GPC by Westonia in the BW population. Another GW QTL was contributed by Spitfire. The second cluster (7B.2) was between 7BM65538 and 7BM74096 (55.0–103.2 cM, 299.2–653.6 Mb) comprising 2559 genes. The QTL of TGW, seed area, seed width, and seed length were contributed by Westonia; and the anthesis, plant height, NUE/PY, by Bethlehem in BW. The Spikelet was attributed to Bethlehem in the SpB population.

One QTL cluster appeared between marker 7DM42766 and 7DM39179 (88.4–111.2 cM, 188.9–548.8 Mb) on 7D, including 2556 genes (Figure 10, Appendix A). The QTL of GY and GW were contributed by Westonia; the anthesis, another GW, biomass, and peduncle length by Spitfire; and the GPC QTL by Bethlehem in the SpB population.

## 3. Discussion

### 3.1. Consensus Map Used for QTL Clusters

Grain yield-related agronomic traits were used as direct selection criteria during wheat breeding based on their high heritability and correlations with grain yield [14]. In the past three decades, molecular markers have been utilized as an efficient tool for molecular marker-assisted breeding and substantially shortened the breeding process. Based on the developed molecular markers, QTL mapping of agronomic traits tends to be a key approach to identifying major QTL and isolating underlying genes in wheat genome [69]. A high-density linkage map and wheat genome sequencing provide more precise genetic information for targeted agronomic traits. The consensus map used in this study is one of ways to enrich the genetic information from different populations [11].

For candidate gene prediction, QTL clusters were identified through a high-density consensus map. A high-density consensus map of BW–SpB was constructed using two DH populations to increase the SNP marker density using LP map R software [69]. The SNP markers were enriched in the overlapping regions, and the gaps between adjacent markers were much smaller in BW–SpB map (Appendix A). The density of BW–SpB map was increased. Compared with the consensus map of Con_map_Wang2014, the higher consistency of marker order showed between the consensus map of BW-SpB and the Chinese Spring physical map. Based on the consensus map of BW–SpB, QTL clusters on chromosomes in two populations were formed.

QTL of agronomic traits in different studies are not always the same due to the differences in the genetic background of materials, variations in environments and cultivation methodologies. Therefore, QTL validation is crucial through replications, multiple environments, and other QTL studies [3]. The robust QTL or QTL clusters in multiple environments provide valuable information for further underlying gene identification.

### 3.2. Confirmation of QTL Clusters and Reported Genes

The QTL of TGW, seed area, seed thickness, seed width, and NUE/PY were grouped in the region of 458.40–494.58 Mb in the physical map, in cluster 1A.1. Similar QTL results of TGW and seed width were reported on this region in previous studies [10,70,71]. Recently, the QTL of grain weight per spike, GN, grain yield (GY), and TGW were also detected in this region [55]. The results indicated that this region contributed TGW and seed-related parameters. Numbers of potential candidate genes are reported in Appendix A. *Glu-1A* (TraesCS1A01G317300; 1A: 508.7 Mb) encoding subunit 1Ax1.1 was next to cluster 1A.1 and may contribute the NUE/PY QTL [72]. Another QTL cluster (1A.2) of GY and NUE/PY was on 1A’s long arm (544.05–591.09 Mb). The QTL of biological yield, plant height, and spike length were also reported to be in this region [55,73]. One QTL of spike compactness was on 1A.2 with an expected candidate gene—TraesCS1A01G425200 (580.4 Mb) [74,75]. *ELF3-1A* (TraesCS1A01G443200; 1A: 591.61) was 0.5 Mb beyond cluster 1A.2. The *ELF3-1A* produces Zinc finger protein CONSTANS, which influences the flowering time. Wheat lines harboring *ELF3* flowered earlier and less spikelets per spike, and stronger photoperiod sensitivity [8,27]. The *ELF3-1A* may contribute to cluster 1A.2.

Phenotypes in three QTL clusters on 1B (10.52–39.46 Mb; 456.27–562.03 Mb; 630.83–681.25 Mb) were slightly different. The phenotypes on 1B.1 included GW, spikelet number, and Keycard; the common phenotypes in clusters 1B.2 and 1B.3 were TGW, height, seed area, seed width and length; and the GN and peduncle length in cluster 1B.3. The QTL of seed length and width, spike length, and spikelet number per spike were identified on 1B’s short arm (8–15 Mb) in a German multi-parental wheat population [73]. The QTL of plant height, TGW, and grain weight per spike were detected between 598 and 641 Mb on 1B [76]. The TGW QTL was also reported on 1B.1 [55] and 1B’s long arm [10]. An *ELF* gene (*TaELF3-1B*) was 4 Mb beyond cluster 1B.3 (685.6 Mb), which may contribute to the QTL. Gene *TaFT3-1B* (1B: 581.4 Mb) was between the QTL clusters 1B.2 and 1B.3. Two other anthesis-related genes, *TaWUSCHELL-1B* (1B: 53.3 Mb) and *TaTOE1-1B* (1B: 59.1 Mb), were about 13-19 Mb beyond 1B.1. It is not clear whether those genes contribute to cluster 1B.1.

There were two QTL clusters on 1D with QTL of biomass, plant height, and NUE/PY in the SpB population on 1D.1 (7.00–32.54 Mb), and the seed width and peduncle ratio on 1D.2 (308.46–411.20 Mb). Similarly, the QTL of flowering time, heading time, and GN were on 1D.1. Heading time and spike weight per spike were in cluster 1D.2 [55]. On the partially overlapped region of 1D.1 (20–49 Mb), the QTL of spike weight and Spikelet were also identified [73]. *TaELF3-1DL* (TraesCS1D01G451200; 1D: 493.4 Mb) showed polymorphisms between Westonia and Bethlehem [8,32]. However, the gene was about 80 Mb beyond cluster 1D.2.

Two QTL clusters were identified on 2A, including the QTL of plant height and seed length (2A.1: 5.36–32.14 Mb), and anthesis, GW, seed area, seed length, Keycard, and peduncle ratio (2A.2: 711.61–757.95 Mb). Correspondingly, the QTL of spike length and spike density were located in cluster 2A.1; and the QTL of GY, peduncle length in cluster 2A.2 [55]. The QTL of spike length was on 31–49 Mb, and the QTL of seed area, seed length and width were in the region of 734–758 Mb on 2A [73]. The QTL of TGW and grain weight per spike (639–733 Mb) partially overlapped with cluster 2A.2 [76]. The grain length QTL was also detected on 2A’s long arm [77]. The *Ppd1-2A* (2A: 36.9 Mb) allele is tightly linked to the QTL cluster 2A.1. *GS2-A2* (TraesCS2A01G500400; 2A: 729.2 Mb) was within the cluster 2A.2 and most likely contributes to the phenotypes [78]. The *GNI1* (grain number increase) gene contributing to GN is on 2A’s long arm (626–631 Mb) [79]. A GA insensitive gene *Rht_NM9* was located between 178.9 and 187.7 Mb on 2AS [42]. *Rht7* was considered as a homoallele of *Rht8* (the collinear chromosome region of *Rht8* was on 3.7 Mb on 2A) [42,80], and the existence of *Rht21* on 2AS was not confirmed [54,81]. Based on the physical map, both *Rht7* and *Rht_NM9* were not within the QTL cluster. One GPC QTL on 2AS (50.0-56.3 MB) without yield penalty was suggested to be used in wheat breeding. A suggested candidate gene, *Fd-GOGAT_2A* (TraesCS2A01G130600; 2A: 78.3 Mb), for a GPC QTL, was 20 Mb beyond the QTL [67,82].

A high number of QTL formed three clusters (2B.1: 44.8–106.00 Mb; 2B.2: 164.09–381.80 Mb; 2B.3: 465.06–721.87 Mb) on 2B. The QTL of GY, GN, GW, plant height, Keycard, and seed length were detected in clusters 2B.1 and 2B.3; the QTL of NUE/PY, seed area were on the 2B.1 and 2B.2 clusters; and the peduncle length QTL was on the 2B.2 and 2B.3 clusters. The QTL of GPC, biomass, and Spikelet were in cluster 2B.1 only. In the study of Corsi et al., three QTL clusters were detected on relevant regions on 2B (33.8–182.0 Mb; 130.3–572.6 Mb; 554.3–648.1 Mb) with phenotypes of flowering time, TGW, seed area, and seed length [73]. Accordingly, there were 38 QTL detected, which formed three QTL clusters (18.9–135 Mb; 143–385 Mb; 554–775 Mb) on 2B based on the physical map in the study of Arif et al. [55]. The phenotypes included GN, peduncle length, plant height, flowering time, heading date, spike density, spike length, and TGW. The QTL of plant height, kernel width, GW per spike, GN, and TGW were also detected on 2B [10,76,77]. It was identified that the copy number differences of *Ppd-B1* strongly influenced the anthesis time in five DH populations [8]. Within QTL cluster (2B.1), a *Ppd1-2B* gene (gene bank number DQ885765) was identified on the short arm of 2B at 56.2 Mb. The QTL of GY, GW, GN, NUE/PY, Biomass, Spikelet, plant height, and Keycard were contributed by Spitfire with a low copy number of *Ppd1-2B*; and GPC by Bethlehem (SpB) with a high copy number, which may strongly contribute to the 2B.1 QTL cluster. *TaSUS2-2B* (TraesCS2B01G194200; 2B: 171.0Mb) was inside of the 2B.2 QTL cluster and may contribute to the phenotypes [83]. *GS2-B2* (TraesCS2B01G528300; 722.6 Mb) was 1 Mb beyond cluster 2B.3 and may contribute to the phenotypes [67,78]. A dwarfing gene *Rht4* was close to a SSR marker WMC317 (4BL: 784.3 Mb) [43], 60 Mb beyond the 2B.3 QTL cluster. Since the precise location of *Rht4* is not clear, it is uncertain if *Rht4* contributes to cluster 2B.3. *Tabas1-B1* (*Tabas1-B1a* and *Tabas1-B1b*)—a nuclear-encoded chloroplast protein [84]—was between Xcfa2278 (2B: 406.5 Mb) and Xbarc167 (2B: 448.7 Mb) on 2BL associated with higher TGW. It is unlikely that *Tabas1-B1* would contribute to cluster 2B.3, since the gene region was 16 MB above cluster 2B.3. *Els2* (early leaf senescence) was tightly linked to the marker 2BIP12 (TraesCS2B01G578800; 767.0 Mb) which was 46 Mb beyond the cluster 2B.3 [85].

Two QTL clusters (320.49–523.15 Mb; 608.20–777.03 Mb) were identified on 2D, including phenotypes of Spikelet and seed length (2D.1), and GPC and seed area (2D.2). Importantly, the GPC QTL stands alone in this region without the negatively pleiotropic effect. One QTL cluster (601–650 Mb) of grain weight per spike, GN, and peduncle length was identified [55], and a TGW QTL was on the 2DL [77]. A QTL of plant height was between markers *Xcfd11* (2D: 79.2 Mb) and *Xgpw361* (106.8 Mb) on 2DS [5]. *GS2-D2* (TraesCS2D01G500600; 595.1 Mb) was 13 Mb above the cluster 2D.2 and may contribute to the GPC QTL [78]. Gene *PpD1-2D* (2D: 33.9 Mb) may not be countable regarding the QTL, as no segregation was identified previously [8]. *TaSUS2-2D* (TraesCS2D01G175600; 2D: 119.0Mb) was outside of the QTL cluster, and may also not contribute to those QTL [83]. One height QTL was also detected between marker 2DM6542 and 2DM5912 (12.7–16.8 Mb) in the SpB population. *Rht8* was physically mapped between 6.3 and 8.1 Mb on 2DS and could increase the spikelet fertility, thereby contributing to increased grain production [42].

On 3A, two clusters were present on 33.19–507.30 Mb (3A.1) and 714.42–717.90 Mb (3A.2). The QTL of anthesis, GN, Spikelet, plant height, peduncle ratio, and seed width were on 3A.1. The QTL of seed area and seed length were in cluster 3A.2. Similar results were reported previously. On cluster 3A.1, the QTL of plant height, peduncle length, spike length, seed number per spikelet, TGW, and heading date (33–418 Mb) were detected; and the QTL of GN, TGW, and GW per spike were located near cluster 3A.2 (691–711 Mb) [55]. The QTL of spike length and Spikelet were in the region of 79–399 Mb; and the QTL of flowering time and seed length were at 665–668 Mb on 3A [73]. The QTL of plant height and GN were found on 3A’s long arm (705–749 Mb) [76]. The dCAPS marker TaTGW6-A1-CAPS-F1 was used to Blast 3A genome, and *TaTGW6-3A* (TraesCS3A01G496900; 3A: 722.4 Mb) was identified, which was close to the 3A.2 QTL cluster. The gene allele *TaTGW6A1a* was associated with higher TGW in comparison with allele *TaTGW6A1b* [86]. *TaGI* (*GIGANTEA*) gene sequences (AF543844) were isolated [87] and were located on 3A (3A: 84.1 Mb). *TaGI-3A* is most likely one of the candidate genes contributing to the 3A.1 QTL cluster. Similarly, *GS5-3A* (3A: 176.5 Mb) blasted by KY661985 was also within 3A.1 [58]. In the 3A long arm, *Eps-3A^m^* (3A: 740.1 Mb) was close to a marker, 3AM70196 (3A: 739.1 Mb), and may not contribute to the 3A.2 QTL cluster [88].

Two QTL clusters were identified on 3B in close regions of 626.09–736.71 Mb (3B.1) and 760.71–809.52 Mb (3B.2). The phenotypes of GW, spikelet number, and seed width were on 3B.1, and GW and NUE/PY were on 3B.2. In the 3B.1 region, Arif et al. identified the QTL of flowering time, heading date, plant height, GN, GW, and spike length on the 3B.2 QTL cluster [55]. Similarly, the QTL of spike weight and GN were detected in the region of 581–692 Mb, and another spike weight QTL was in the region of 796–813 Mb on 3B [73]. *TaTGW6-3B* showed three copies within 3B.2 (3B: 793019222–793062656), TraesCS3B01G558800, TraesCS3B01G559000, and TraesCS3B01G559100. These genes may contribute to those QTL. *Rht5* was mapped close to SSR marker Xbarc102 on 3BS, whereas the SSR marker sequences failed to successfully Blast the physical map location.

Two QTL clusters were located on 4A (4A.1: 2.99–16.71 Mb, 4A.2: 601.23–674.86 Mb). The 4A.1 cluster included GY, GW, GN, seed area, and seed thickness; and 4A.2 contained TGW, plant height, peduncle length, peduncle ratio, seed area, seed width and length. It seems that on the short arm, GY derived from GN; and on the long arm, plant height was associated with TGW. Similarly, the QTL of plant height, flowering date, heading date, Spikelet, and spike length were on the region of 594–743 Mb on 4A [3]; and the QTL of TGW, GN, and GW were in the region of 622–685 Mb on 4A [76]. The QTL of GN, GW, seed number per spikelet, spike length, plant height, flowering time, and heading date were detected on the 4A.2 region [55]. Neighboring 4A.2, the QTL of spike length, spike weight, and spikelet per spike were in the region of 545–601 Mb on 4A [73]. *WSOC1-4A* (TraesCS4A01G320300; 4A: 608.8 Mb), a homologous gene sequence of *WSOC1-4D* was within cluster 4A.2 may contribute to the QTL, while the homologous gene of the DELLA protein (*rht1-D1a*; AJ242531)-TraesCS4A01G271000 (4A: 582.4 Mb) [89], is close to 4A.2 QTL cluster.

Two QTL clusters were formed on 4B (4B.1: 1.26–52.07 Mb, 4B.2: 574.95–665.06 Mb) with the QTL of GY and anthesis on 4B.1; and the QTL of anthesis, peduncle length, peduncle ratio, TGW, seed area, seed thickness, and seed width on 4B.2. In the study of Corsi et al., the QTL of plant height and spike length were detected in the region of 31–165 Mb on 4B [73]. The TGW QTL was also found on the 4BS [77]. On cluster 4B.2, the QTL of GN, seed number per spikelet, spike length, plant height, TGW, and spike density were located in this region [55]. The QTL of plant height was also detected between 663 and 666 Mb on 4B [3]. Since all three parental lines of BW and SpB populations harbor *RhtD1a* and *RhtB1b*, no segregation for *Rht1* and *Rht2* genes was presented in those two populations. Gene *WSOC1-4B* (TraesCS4B01G346700; 4B: 640.3 Mb) within the cluster 4B.2 may contribute to the phenotypes.

A QTL cluster of GPC and TGW was in the region of 30.02–150.00 Mb in the physical map on 4D. Likewise, the QTL of TGW, GN, GW per spike, flowering time, and heading data showed in this cluster [55]; and the QTL of TGW, plant height, and seed area were also located at 3–108 Mb on 4D [73]—similar to the results of the QTL of plant height and TGW (4D: 12–62 Mb) [76]. As mentioned above, *Rht2* was not segregating in those two populations, and *WSOC1-4D* (TraesCS4D01G341700; 4D: 498.3 Mb) was outside of this QTL cluster. The potential candidate genes in 4D cluster are suggested in Appendix A Appendix A. There is agronomic benefit to identify the genes contributing to this cluster in those two populations, as no GY penalty was noticed for GPC and TGW. *TaGS1-4D*, a glutamine synthetase gene Blasted by AY491970 (TraesCS4D01G047400, 4D: 22.9 Mb), was 7 Mb above the cluster [90].

A high number of QTL were detected on 5A, especially in the SpB population, and formed three QTL clusters at 441.02–467.38 (5A.1), 515.92–549.36 (5A.2), and 579.05–653.86 Mb (5A.3). On cluster 5A.1, the QTL included Keycard, NUE/PY, plant height, Spikelet, and seed width; and in clusters 5A.2 and 5A.3, 14 QTL were included in the SpB population, such as anthesis, GN, GPC GW, plant height, Keycard, NUE/PY, peduncle length, TGW, seed area, seed length, seed thickness, seed width, and spike length. In the study of Arif et al., the QTL of plant height, spike length, peduncle length, and grain yield were on 5A.1; and the QTL of GW, spike length, plant height, peduncle length, TGW, heading date, flowering time, grain yield, and spike density were in clusters 5A.2 and 5A.3 [55]. Correspondingly, the QTL of flowering date, heading date, plant height, spike length, and Spikelet were detected between 424 and 476 Mb and the QTL of flowering date, spikelet per spike, and spike length were at 570–613 Mb on 5A [3]. The QTL of flowering time was in the region of 547–552 Mb on 5A [73]; and the QTL of GN, TGW, and GW were in the region of 11–460 Mb on 5A [76]. The TGW QTL was also reported to be on 5AL [77]. *VrnA1a* (TraesCS5A01G391700; 5A: 587.4 Mb) harbored by Bethlehem was segregating in the SpB population (marker VPA) within cluster 5A.3. The pleiotropic effects of *VrnA1a* are clear within this region, as Spitfire possesses the winter-type of *VrnA1b* contributing to the GN and days to anthesis; other phenotypes attributed to Bethlehem within the cluster. *VRN2* (*ZCCT1*) (TraesCS5A01G541200; 5A: 698.1 Mb) is located at the end of 5A [17], which is out of the QTL cluster region. *Rht12* was close to a SSR marker of W5AC207-5A (5A: 698.8 Mb) which was 40 Mb beyond 5A.3 QTL cluster [91]. *Rht9* (close to SSR marker BARC151, 5A: 560.1 Mb), and a grain length gene *TaGl3-5A* (TraesCS5A01G373900; 5A: 571.78 Mb) was next to cluster 5A.3 and may also contribute the seed length QTL [43,61].

There were two QTL clusters on 5B (368.50–533.03 Mb (5B.1) and 613.61–701.46 Mb (5B.2)). Cluster 5B.1 included Spikelet, Keycard, seed area, seed width and thickness. The phenotypes in cluster 5B.2 were anthesis, GPC, NUE/PY, GW, TGW, seed area, seed thickness, and seed width. Interestingly, in cluster 5B.2 one GPC QTL was parallel with the days to anthesis QTL, and both were contributed by Spitfire. Similarly, the QTL of flowering time, heading date, peduncle length, plant height, grain yield, and GN were in cluster 5B.1; and the QTL of GN, flowering time, heading date, and spike length were in cluster 5B.2 [55]. The QTL of flowering date, plant height, and Spikelet were in the region of 559–682 Mb on 5B [3]. The QTL of spike weight, GN, spike length, and flowering time were in the region of 4–539 Mb; and the QTL of spike length were also detected in the region of 683–700 Mb on 5B [73]. *VrnB1a* (TraesCS5B01G396600; 5B: 573.8 Mb) was segregating in both BW and SpB populations (marker VrnB). Interestingly, the gene marker was between the two QTL clusters. A homologous gene to *TaGl3-5A*, *TaGl3-5B* (TraesCS5B01G375800; 5B: 553.34 Mb), was 20 Mb beyond cluster 5B.1, and may not contribute to the seed parameters in this cluster [61].

The QTL of GW, anthesis, and seed width formed one QTL cluster at 338.17–467.18 Mb on 5D. In the study of Arif et al., the QTL of peduncle length, plant height, GN, and GW were detected in this region [55]. *VrnD1a* (TraesCS5D01G401500; 5D: 467.17 Mb) possessed by Bethlehem was in the QTL cluster region and segregating in the BW population [8]. The GW and days to anthesis were contributed by winter-type *VrnD1b* harbored by Westonia, and the seed width by Bethlehem possessing *VrnD1a*. A homologous gene to *TaGl3-5A*, *TaGl3-5D* (TraesCS5D01G383300; 5D: 452.84 Mb), was inside of the QTL cluster on 5D, and may contribute to the seed width in this cluster [61]. *Rht23* was between the SSR markers Gdm63 and Barc110 on 5DL (5D: 517.45–529.26 Mb) and showed dwarfness and compact spikes [52]. *Rht23* may contribute to the QTL of spikelet per spike and peduncle ratio (5D: 488.82–547.12 Mb) outside of 5D QTL cluster in the current study. *VRN-D4* on 5DS is a homologous gene to *Vrn-A1* [19]. *VRN-D4* was segregating in the BW population [8]. Based on primer blast, *VRN-D4* (5D: 174.3 Mb) is outside of this QTL cluster region, but inside a region of GW QTL between markers 5DM62708 and 5DM53180 (5D: 161.66–407.97 Mb) on 5D in the current study.

QTL of plant height, seed area, and seed width were grouped in one QTL cluster on the top of 6A (0.96–13.81 Mb). Accordingly, the TGW QTL was on this cluster on 6A [55]. The QTL of spike weight was in the region of 5–6 Mb; and the QTL of plant height, TGW, flowering time, and spike weight were in the region of 455–611 Mb on 6A [73]. Guan et al. stated that the QTL of plant height, TGW, and GN were over a more broad region (6A: 38–596 Mb) [76]. A few GA responsive genes were identified on 6A. *Rht14* was mapped in the region 383–422 Mb [92]. GA-responsive gene *Rht18* (gibberellin-2-oxidase-A9) (TraesCS6A01G221900; 6A: 413.7 Mb) on 6A moderately reduced the plant height and increased the harvest index [93]. *Rht24* was mapped in the same region as *Rht14*, *Rht16*, and *Rht18*, whereas *Rht24* showed higher frequency in European and Chinese wheat cultivars [53]. It is not clear whether these four genes are alleles of the same gene or different genes from a *Rht* gene cluster [53,94]. *Rht25* was identified at 144.0–148.3 Mb on 6AS and was different from *Rht18* [47]. All *Rht* genes above are not within the QTL cluster region. *TaGW2-6A* (TraesCS6A01G189300; 6A: 237.7 Mb) was associated with grain weight [56]. *TaGW2-6A* and *Rht18* (*Rht14*, *Rht16 and Rht24*) may contribute to the QTL of Spikelet (200.56–459.85 Mb) detected in the current study. Flowering related gene-*TaCO-A2* (GenBank number: MT043304) (TraesCS6A01G289400; 6A: 521.4 Mb) was out of the QTL cluster [34].

Two QTL clusters were clearly formed in the regions of 48.91–138.80 Mb (7A.1) and 696.98–730.32 Mb (7A.2) on 7A, including 10 and five phenotypes, respectively. In cluster 7A.1, the QTL included GY, GW, GPC, TGW, GN, biomass, seed area, seed width, seed length, and seed thickness; and cluster 7A.2 included GY, GW, GPC, seed width, and GN. In recent QTL studies, the QTL of TGW, heading date, spike length, and GW were in the 7A.1 region; and the QTL of plant height, GN, GW, TGW, and grain yield were in cluster 7A.2 on 7A [55]. Likewise, the QTL of spike length, seed area, spike weight, TGW, flowering time were in the region of 41–112 Mb while the QTL of spike length, Spikelet and GN were in the region of 675–736 Mb on 7A [3,73,76]. Several grain weight-related genes were on 7A. *vrn-A3* (TraesCS7A01G115400; 7A:71.6 Mb) [95] was in cluster 7A.1 QTL, similarly to *TaSUS1-7A* (TraesCS7A01G158900; 7A: 115.2 Mb) [83] and *TaVrt2* (TraesCS7A01G175200; 7A: 128.8 Mb) [20,21]. Those genes most likely contribute to the QTL in 7A.1. Other genes reported were not inside of the QTL cluster in the current study, for example, *TaGASR7-A1* (TraesCS7A01G208100, 170.6 Mb bp) was associated with grain weight [96] and *TaCO-A1* (TraesCS7A01G211300; 7A: 174.2 Mb) interacted with *PPD1* to regulate photoperiodic response [34]. *TaTGW6-7A* (TraesCS7A01G233600; 7A: 205.4 Mb) was reported, and *TaTGW6-7Aa* were associated with higher TGW [60]. *TaSAP1-A1* (7A: 585.0 Mb) was associated with TGW and GN, spike length peduncle length, and Spikelets [97]. *WAPO-A1* (TraesCS7A01G481600; 7A: 764.0 Mb) contributed to the QTL of Spikelet on 7AL [6]. A dwarf gene *Rht22* was located between SSR markers Xgwm471 and Xgwm350 (1.46–3.20 Mb) on 7AS [54], outside of the clusters.

Two QTL clusters were identified in the regions of 1.80–16.83 Mb (7B.1) and 299.25–653.57 Mb (7B.2) on 7B. The three phenotypes of GY, GW, and GPC were in cluster 7B.1 and eight phenotypes of anthesis, plant height, TGW, NUE/PY, Spikelet, seed area, seed width, and seed length in cluster 7B.2. Correspondingly, the QTL of flowering time, heading date, GW, and spike length were in cluster 7B.1; and the QTL of spike length and plant height were in cluster 7B.2 [55]. Similarly, a flowering date QTL was at 15–41 Mb and the QTL of flowering date, heading date, and spikelet per spike were in the regions of 545–593 Mb on 7B [3]. The QTL of spikelet number per spike was in the region of 472–667 Mb on 7B [73]. *vrn-B3* (TraesCS7B01G013100; 7B: 9.7 Mb) is within cluster 7B.1 and may contribute to the QTL [95], whereas *TaVrt2* (TraesCS7B01G080300; 7B: 90.18 Mb) was beyond cluster 7B.1 [20,21]. *TaSAP1-7B* (7B: 544.4 Mb) within 7B.2 [97] may be attributable to the cluster. *TaTGW6-7B* (TraesCS7B01G131900; 7B:159.5 Mb), which was orthologous to *TaTGW6-7A,* was not in the region of clusters [60], similarly to *TaCO-B1* (TraesCS7B01G118300; 7B: 137.7 Mb) [34].

Probably due to fewer SNPs, one QTL cluster was formed with a relatively large region of 188.87–548.80 Mb in the physical map on 7D. The QTL included GY, GW, anthesis, GPC, biomass, and peduncle length. Accordingly, the QTL of flowering time and test weight were detected in this region [55]. Corsi et al. detected the QTL of spike weight and GN in the region of 13–16 Mb on 7D [73]. Likewise, one GY QTL was at 16.4–18.4 Mb on 7D in the current study. *TaTGW6-7D* (TraesCS7D01G233800; 7D: 195.6 Mb), orthologous to *TaTGW6-7A*, was within the QTL cluster [60], similarly to *TaSAP1-7D* (7D: 512.2 Mb) [97]. These two genes may contribute to the QTL. *TaCO-D1* (TraesCS7D01G212900; 7D: 171.3 Mb) was 18 Mb ahead of the cluster [34]. The following three genes were out of the QTL regions: *TaGS-D1* (TraesCS7D01G015000l; 7D: 6.4 Mb bp) associated with higher TGW and grain length [98], *vrn-D3* (TraesCS7D01G111600; 7D: 68.4 Mb) [95], and *TaVrt2* (TraesCS7D01G176700; 7D: 128.91 Mb) [20,21].

### 3.3. Trait Correlations within QTL Clusters and Applications in Breeding

From the average data of each population in each location, the large environmental variations were shown. The highly stable phenotypes (*h^2^* > 0.85 in both populations) were days to anthesis, TGW, Spikelet, seed area, seed width, and seed length. The stability of seed length was also observed by previous studies [77,99].

Expectedly, significantly positive correlations appeared between grain yield and yield components, such as, GY, NUE/PY, biomass, GW, Spikelet, and GN, in both populations, and two or more environments. GY also showed significantly positive correlations with plant height and days to anthesis. The TGW showed significantly positive correlations with seed parameters, plant height, stem length, peduncle length, and spike length. Unexpectedly, the positive correlation between TGW and GY was only shown in the SpB population in Muresk. The GPC expectedly had significantly negative correlations with GY, grain yield components, and days to anthesis in both populations and all environments, and positive correlations with TGW and seed parameters in BW in Williams. The GN was also negatively associated with the TGW and seed related parameters, but positively associated with GY, days to anthesis, and plant height. The results indicate that high plants had high TGW and GN, leading to high grain yield. The negative correlations between peduncle ratio and phenotypes of spike length, stem length, NUE/PY, GY, and grain yield components indicate that a low proportion of peduncle predicts a high grain yield. This study revealed the positive correlations between the Keycard and the traits of the spikelet number per spike and GY. The result indicates that the greater Keycard may be a favorable phenotype for higher GY selection.

Although the correlations above provide indications for selection, the complex correlations between traits also cause difficulties for combining desired traits in wheat breeding. For example, TGW is usually negatively associated with GN; both traits contributed to grain yield. A certain level of TGW is required for wheat grain. GPC tends to have a negative correlation with grain yield, whereas high GPC is a desirable trait for grain quality. Profoundly, the traits in QTL clusters revealed the trait linkages and more precisely showed the alignments between traits along with chromosome locations and markers, which would provide a track for selections in wheat breeding (Appendix A), as optimizing yield-related gene combinations is an essential genetic approach for improving grain yield and grain quality through wheat breeding [100].

From the correlation analysis, TGW was positively associated with seed parameters and plant height. In the QTL clusters, the QTL of TGW aligned with seed area on 1A.1, 1B.2, 1B.3, 4A.2, 4B.2, 5A.3, 5B.2, 7A.1, and 7B.2; with plant height on 1B.2, 4A.2 and 5A.3. The TGW QTL was negatively associated with GN on 1B.3 and 5A.3; and with grain weight and NUE/PY on 1A.1, 5B.2, and 7B.2. There were six TGW QTL without negative impacts—1B.2, 4A.2, 4B.2, two on 4D, and 7A.1.

The negative correlations between GPC and grain yield, including GW and NUE/PY, appeared on QTL clusters 2B.1, 2B (767.6–770.5 Mb), 5B.2 (612.1–613.6 Mb), 7A.1, 7A.2, 7B.1, and 7D. Most contributions were from Bethlehem, except the 7B.1 from Westonia. There were five QTL of GPC without yield penalty, which were on 2A (50.0–56.3 Mb, by Westonia), 2D and 4D (by Bethlehem), 5B.2 (671.3–672.9 Mb, by Spitfire), and 6B (by Westonia).

The GN showed positive correlations with grain yield. Nevertheless, the negative correlations were present in several QTL clusters. The GN QTL aligned positively with the QTL of grain yield in cluster 2B.3, without having negative impacts on other traits on QTL clusters 3A.1, 4A.1, and 7B (712.3–717.3 Mb). The GN QTL showed negative impacts on TGW on 1B.3; on GW, NEU/PY, and TGW on QTL cluster 5A.3.

The days to anthesis showed positive correlations with grain yield in correlation analysis. However, both positive and negative linkages appeared between days to anthesis and GY within QTL clusters. The QTL of days to anthesis aligned positively with GN on 5A.3, with NUE/PY on 5B.2 (671.3–692.9 Mb) and 7B.2, and with GW on 5D and 7D; but negatively with NUE/PY, GW, and TGW in clusters 2A.2 and 5A.3, with TGW on 4B.2 and 7B.2, and with GPC on 7D. The anthesis QTL on 4B.1 did not have strong impacts on other traits.

The Keycard showed positive correlations with GW, Biomass, NUE/PY, and spikelet number per spike. The alignments were shown for clusters 1B.1, 2A.2, 2B.1, 3D, 5A.1, 5A.3, and 5B.1.

Plant height was positively associated with GW and NUE/PY, and the alignments were shown for clusters 2B (linkage: 4.9–16.5 Mb), 2B.1, 3B (linkage: 68.9–214.5 Mb), 5A.2, 5A.3, and 7B.2.

The peduncle ratio was negatively associated with GW and spikelet number per spike, which indicated that the selection with low peduncle ratio may result in high grain yield. The negative alignments were in clusters 2A.2, 3A.1, the linkage on 3B (68.9–214.5 Mb), and 5D (448.8–547.1 Mb).

The current study indicates that the QTL clusters with consistently positive correlations between yield-related traits could be used in wheat breeding selection, such as, 1B.2, 2A.2, 2B (linkage: 4.9–16.5 Mb), 2B.3, 3B (linkage: 68.9–214.5 Mb), 4A.2, 4B.2, 4D, 5A.1, 5A.2, 5B.1, and 5D (with wildtype *VrnD1b*). The negative alignments require more consideration for the trait selections on the QTL clusters of 1A.1, 2B.1, 1B.3, 5A.3, 5B.2 (612.1–613.6 Mb), 7A.1, 7A.2, 7B.1, and 7B.2. One GPC QTL (5B.2: 671.3–672.9 Mb) contributed by Spitfire was positively associated with NUE/PY and could be used in selection; and the GPC QTL without negatively pleiotropic effects on 2A (50.0–56.3 Mb, by Westonia), 2D, 4D, and 6B were suggested to be used in high quality wheat breeding.

In summary, two DH populations were planted at four different locations in two years with two replicates in each location. Fine maps and consensus maps were constructed according to 12,000 GBS SNP markers. Overall, 22 traits were evaluated, and 227 QTL were detected after combining several traits, different environments, and neighboring individual QTL. The QTL on 18 chromosomes formed 35 QTL clusters. Potential candidate genes in each QTL cluster were predicted. The associated QTL results and candidate genes identified by previous researchers were corroborated. The QTL clusters with negative linkages between TGW and GN were identified, along with the negative linkages between GPC and GY, and between TGW and GW. The QTL clusters with positive linkages between traits or without negative impact were suggested to be used in wheat breeding. This study may provide new insights of specific genome regions for major gene prediction.

## 4. Materials and Methods

### 4.1. Plant Materials

This study was conducted using two DH populations derived from the following crosses: Bethlehem/Westonia (BW) and Spitfire/Bethlehem (SpB). Spitfire and Westonia are mid to early flowering varieties and produce good yield. Bethlehem flowers early and produces medium yield. All three varieties harbor *RhtD1b* and wildtype of *RhtB1a*. Therefore, there is no segregation for *Rht1* and *Rht2* genes. However, those three parental lines hold different alleles of *VRN1*, as Bethlehem possesses *Vrn-A1a and Vrn-D1a*, whereas Westonia harbors *Vrn-A1a* and *Vrn-B1a*, and Spitfire contains *Vrn-B1a* and *Vrn-D1a*. Therefore, the *Vrn-B1a* and *VrnD1a* segregated in the BW population; and *Vrn-A1a* and *Vrn-B1a* in the SpB population. The two DH population sizes were 105 and 168 for BW and SpB, respectively [8].

### 4.2. Field and Glasshouse Experiments

In 2017, individual lines from those two DH populations were planted as 1-m^2^ plots in Katanning (Kat), Wongan Hills (WH), and South Perth, and one line per pot (4 L) in a glasshouse (GH) at Murdoch University. The QTL results of the flowering data in these trials were included in this study.

In 2018, the field experiments with the two DH populations were conducted at three locations across Australia representing distinct environments, including Narrabri in New South Wales, Muresk, and Williams in Western Australia (WA). The majority of the DH lines (>96%) were replicated two times at one or more than one of the three locations. Parental lines were also utilized in each of the field experiments. The partially replicated experiments were designed using DiGGer in R [101]. The harvested plot sizes were 3.3 m^2^ (1.65 m × 2 m), 4.6 m^2^ (1.15 m × 4 m), and 5.9 m^2^ (1.3 m × 4.5 m) in Narrabri, Muresk, and Williams, respectively (Appendix A). Based on the TGW of individual lines, about 625 seeds were sown in each plot (5–7.8 m^2^). In 2020, a similar field experiment was conducted in Beverley, WA. The plot size was 7.2 m^2^ (1.2 m × 6 m). Two replicates of each DH line were randomly planted based on the DiGGer design (Appendix A). The harvested area of each plot was 4.8 m^2^ (1.2 m × 4 m). The number of plots of BW population were 176, 192, 240, and 192 in Narrabri, Muresk, Williams, and Beverley, respectively; and the plots numbered 264, 288, 360, and 240 for the SpB population (Appendix A).

Field-grown plants were rainfed under standard agronomic management practices, whereas adequate water and fertilizer were provided to plants grown in pots in the glasshouse (2017) [102].

### 4.3. Core Phenotype Measurements

Anthesis time was recorded for the most of field trials. Zadok stages around 50–60 were recorded in Narrabri field trial. Maturity time was recorded for the trials in Muresk and Williams only. The calculations for the days to anthesis were described previously [8]. In field trials in Muresk and Williams, plant number was randomly recorded at 2 × 1 m length in each plot at the tillering stage. Apart from machine harvesting, 1 m^2^ of plants in each plot was hand-harvested for detailed phenotype recording. The spike number was counted before hand-harvesting. The total biomass was weighed and recorded before threshing. The grain weight per spike and grain number per spike were calculated based on the spike number per m^2^ after threshing. The grain weight per plant was calculated based on the plant number per m^2^. Since only Williams plants had enough tillers, the grain weight per plant was used in the correlation analysis and QTL analysis in the Williams trial only. Before harvesting, plant height, spike length, stem height, and peduncle length were recorded using three main tillers of three representable plants in each plot. The peduncle ratio was calculated as peduncle length/stem length ×100. In addition, in Muresk, Williams, and Narrabri, 10 spikes of each plot were collected for spikelet number per spike and keycard dimeter recording. After harvesting, the GPC, TGW, and seed parameters— such as, seed length, seed width, seed thickness, and seed area—were measured using NIR seed scanner and seed count machine (SeedCount SC6000R analyser from Next Instruments Ltd., Australia). The grain yield (GY) was recorded, and the protein yield (PY) and nitrogen use efficiency (NUE) were calculated as follows: PY = GY × GPC; NUE = PY/total nitrogen applied (soil nitrogen was included).

### 4.4. Linkage Map Construction

The map construction was described previously [8]. In short, genomic DNA was extracted from a single plant for each DH line and the parental lines [103], and 12k tGBS SNP was used for linkage map constructions. Identical lines were detected and removed using non-metric multidimensional scaling (MDS) of genetic dissimilarity using software from Numerical Taxonomy System (NTsys) v2.2 and Plymouth Routines in Multivariate Ecological Research (PRIMER v6) [104,105]. Lines with large proportions of missing values during SNP genotyping were also removed, together with distorted markers and double-cross markers. Most co-segregating markers were made redundant and removed from the genetic map. As a result, a fine map of BW population was constructed using 77 lines and 2387 SNP markers; and SpB by 94 DH lines and 2570 SNP markers (Appendix A). In addition, markers for *Vrn-A1a*, *Vrn-B1a*, and *Vrn-D1a* were used to construct the final maps of the two DH populations using Map Manager [106] and the QTL mapping package R/qtl [107]. Primers for *VRN1* (*Vrn-A1a*, *Vrn-B1a*, and *Vrn-D1a*) were as described [108].

LPmerge software was used to construct a consensus map with calculation the consensus marker orders of linkage groups based on the individual maps [69]. The consecutive curves and the circus graph of collinearity among genetic maps were created through the circlize package in R.

### 4.5. QTL Mapping

Inclusive composite interval mapping (ICIM) (IciMapping V4.1; http://www.isbreeding.net, accessed on 1 September 2016) was employed for QTL detection [109]. The value of each replicate and the mean value of replicates of individual DH lines in each environment were used for QTL detection under a LOD score of 2.5. Permutations were set to 1000 at a significance level of 0.05. The inclusive composite interval mapping addition (ICIM-ADD) method was selected for QTL mapping [110].

### 4.6. Statistical Analysis

In the analysis of the phenotypic traits, linear mixed models (LMM) were fitted with ASReml-R [111], where the variance parameters in the mixed model were estimated using the residual maximum likelihood (REML) procedure [112]. Residual diagnostics were used to examine the validity of the model assumption (normality and homogeneity of variance). The linear unbiased predictions (BLUPs) were employed for the phenotypic traits.

Phenotypic data were analyzed by multivariate analysis of variance (MANOVA) using the general linear model implemented in IBM SPSS statistics 24 (https://www.ibm.com/au-en/products/spss-statistics). Post hoc Tukey’s multiple range tests were used to identify significant groupings under the same environment or across different environments. The box and violin plots were generated through the ggplot2 package in R. The broad-sense heritability was calculated through the package lme4 in R-studio. The pairwise Pearson correlations of the parameters were computed using the “rcorr” function implemented in the R package Hmisc [113] using the BLUP values across environments.

## Figures and Tables

**Figure 1 ijms-22-11934-f001:**
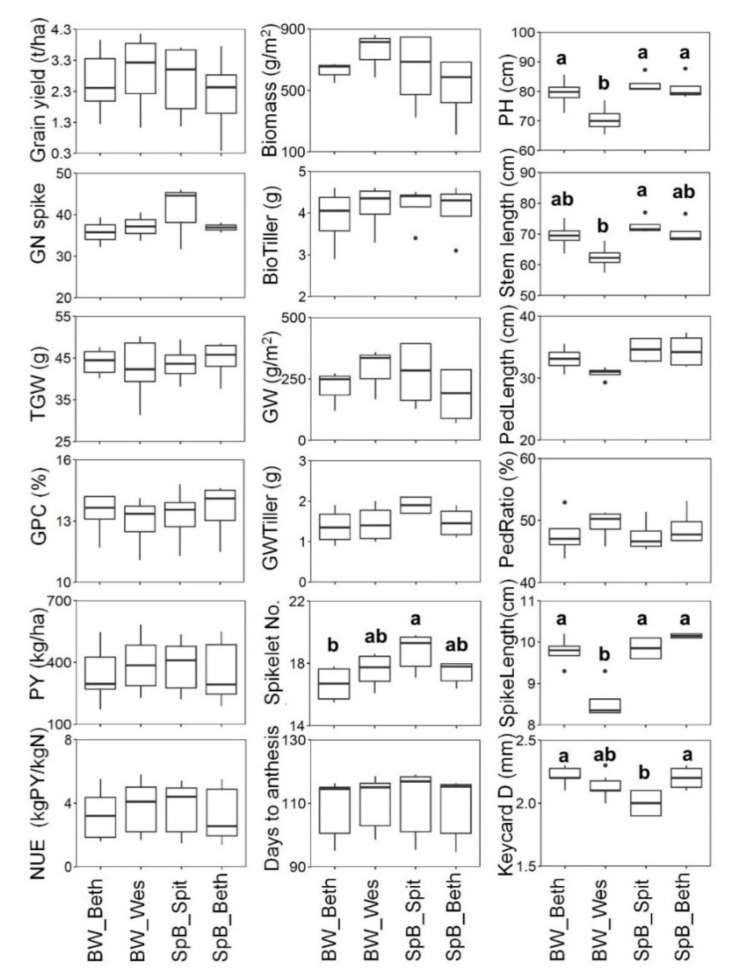
Box plots of yield-related traits of three parental lines in BW and SpB populations. GN: grain number; TGW: thousand grain weight; GPC: grain protein content; PY: protein yield; NUE: nitrogen use efficiency; Biomass: plant biomass per m^2^; BioTiller: biomass per tiller; GW: grain weight; GWTiller: grain weight per tiller; Spikelet No.: spikelet number per spike; PH: plant height; PedLength: peduncle length; PedRatio: peduncle ratio; Keycard D: keycard diameter; Beth: Bethlehem; Wes: Westonia; Spit: Spitfire. Values with the same letter or without letter are statistically not different at *p* = 0.05.

**Figure 2 ijms-22-11934-f002:**
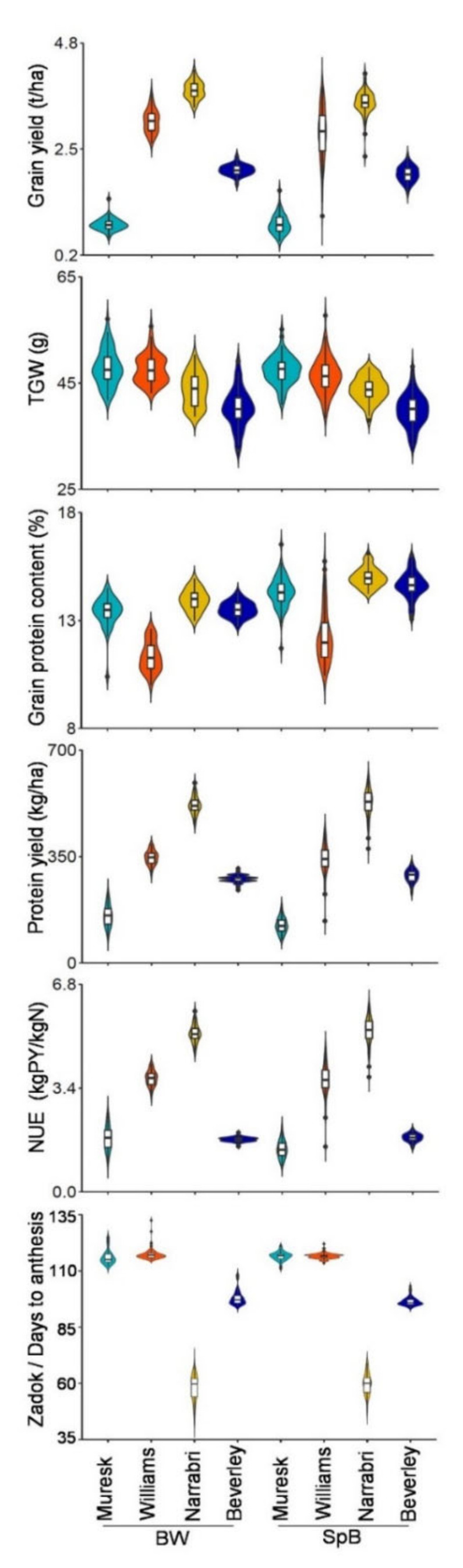
Violin plots of yield-related traits and grain protein content of BW and SpB populations. TGW: thousand grain weight; NUE: nitrogen use efficiency.

**Figure 3 ijms-22-11934-f003:**
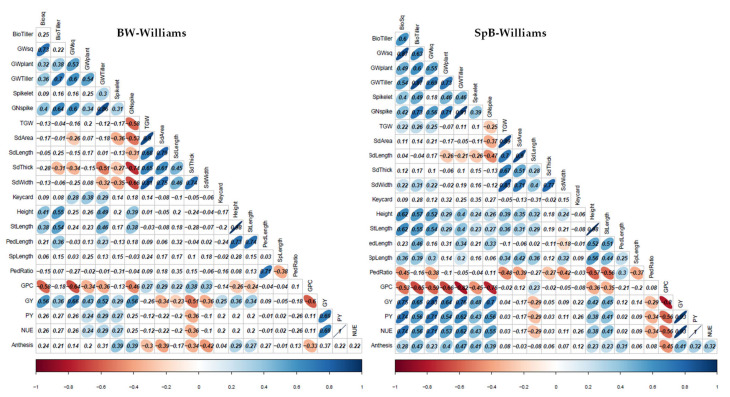
Phenotypic correlations between yield-related traits in Williams in BW and SpB populations. The values are correlation coefficients r. The areas and colors of ellipses show the absolute values of the corresponding correlations. Right and left oblique ellipses indicate positive and negative correlations, respectively. The values without ellipses indicate insignificance at the 0.05 level. Biosq: plant biomass per m^2^; BioTiller: biomass per tiller; GWsq: grain weight per m^2^; GWplant: grain weight per plant; GWTiller: grain weight per tiller; Spikelet: spikelet number per spike; GNspike: grain number per spike; TGW: thousand grain weight; SdArea: seed area; SdLength: seed length; SdThick: seed thickness; SdWidth: seed width; Keycard: keycard diameter; Height: plant height; StLength: stem length; PedLength: peduncle length; SpLength: spike length; PedRatio: peduncle ratio; GPC: grain protein content; GY: grain yield; PY: protein yield; NUE: nitrogen use efficiency; Anthesis: days to anthesis.

**Figure 4 ijms-22-11934-f004:**
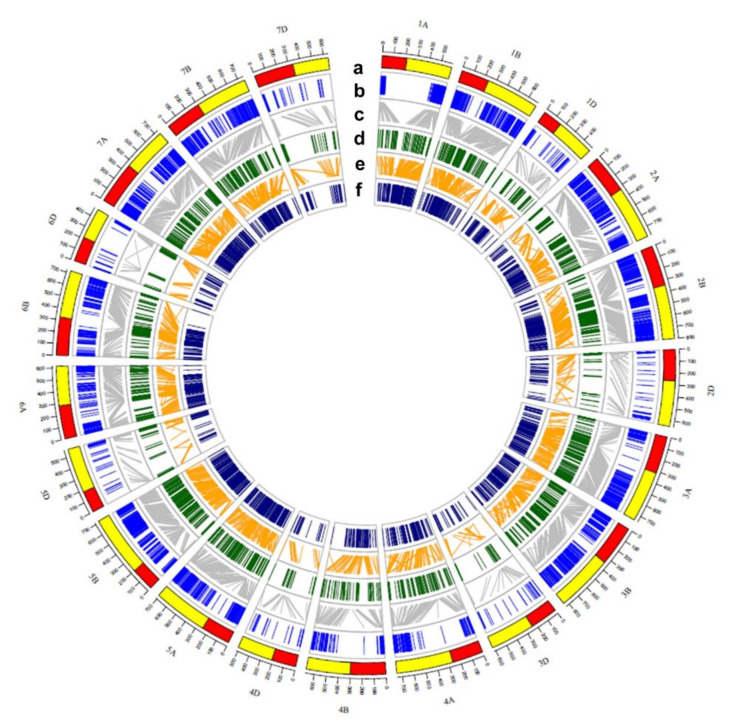
The circos graph of the consensus map of BW–SpB, the consensus map of Wang et al. (2014), and the physical map of the wheat genome (IWGSC RefSeq v1.0). (**a**) The chromosomes and the physical location scale of wheat. (**b**) Physical map of the SNP markers using in BW–SpB. (**c**) The collinearity between the consensus map of BW–SpB and the physical map. (**d**) The consensus map of BW–SpB. (**e**) The collinearity between BW–SpB and the consensus map of Wang et al. (f) The consensus map of Wang et al. (2014).

**Figure 5 ijms-22-11934-f005:**
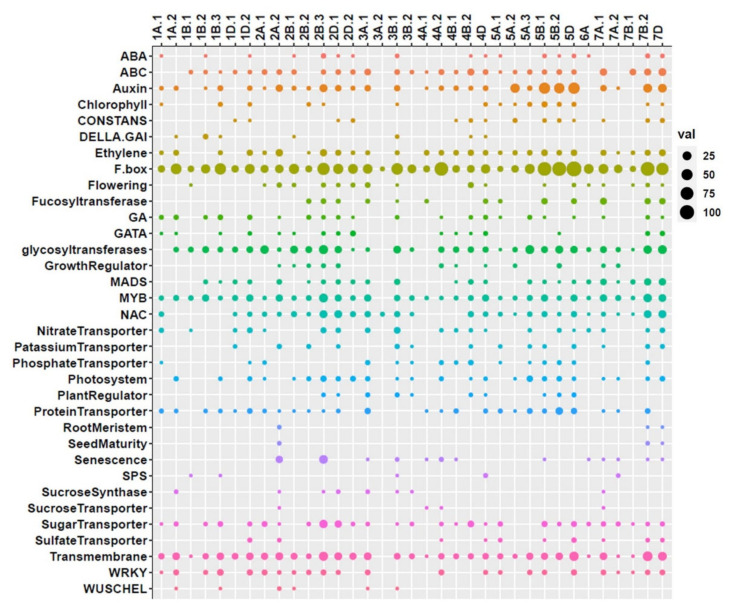
Potential candidate genes in QTL clusters in 18 wheat chromosomes. ABA: abscisic acid; ABC: ABC transporter ATP-binding protein ARB1; GA: gibberellin acid; GATA: GATA transcription factor; MADS: MADS-box transcription factors; MYB: MYB related transcription factors; NAC: NAC domain proteins; SPS: sucrose phosphate synthase; WRKY: WRKY transcription factors; WUSCHEL: *WUSCHEL-like* homeobox.

**Figure 6 ijms-22-11934-f006:**
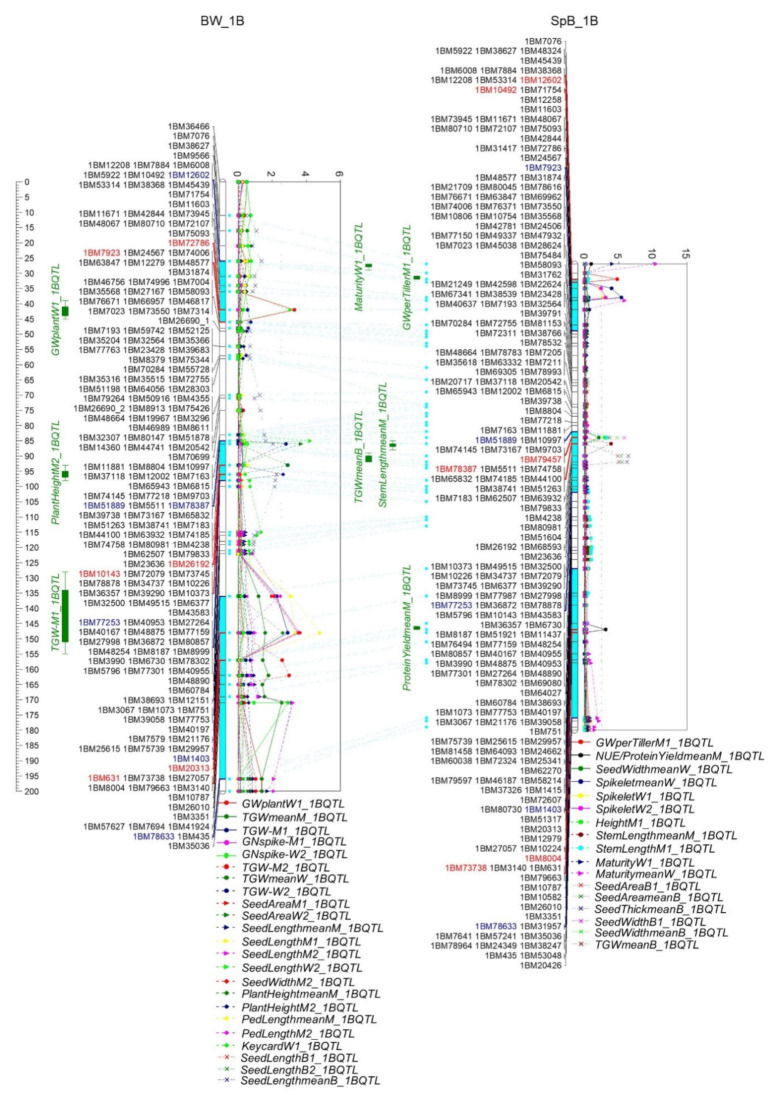
Significant QTL formed three QTL clusters on the homologous region on 1B in both populations, BW and SpB. Homologous regions in both populations are highlighted in light blue; markers for QTL highlighted in red; homologous markers highlighted in blue.

**Figure 7 ijms-22-11934-f007:**
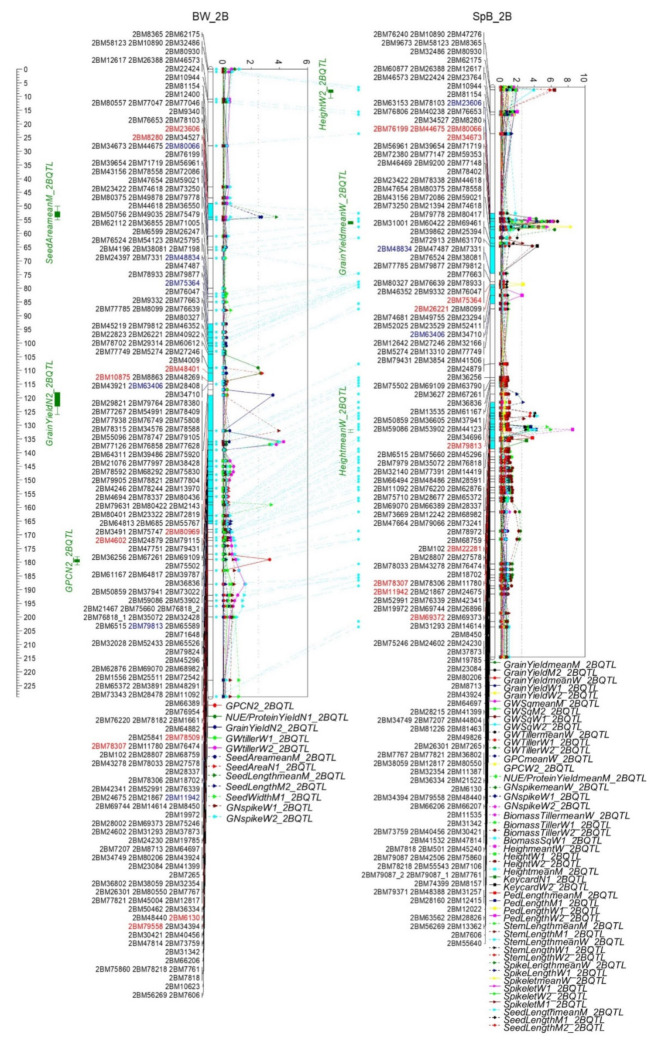
Significant QTL formed three QTL clusters on the homologous regions on 2B in BW and SpB populations. Homologous regions in both populations are highlighted in light blue; markers for QTL highlighted in red; homologous markers in blue.

**Figure 8 ijms-22-11934-f008:**
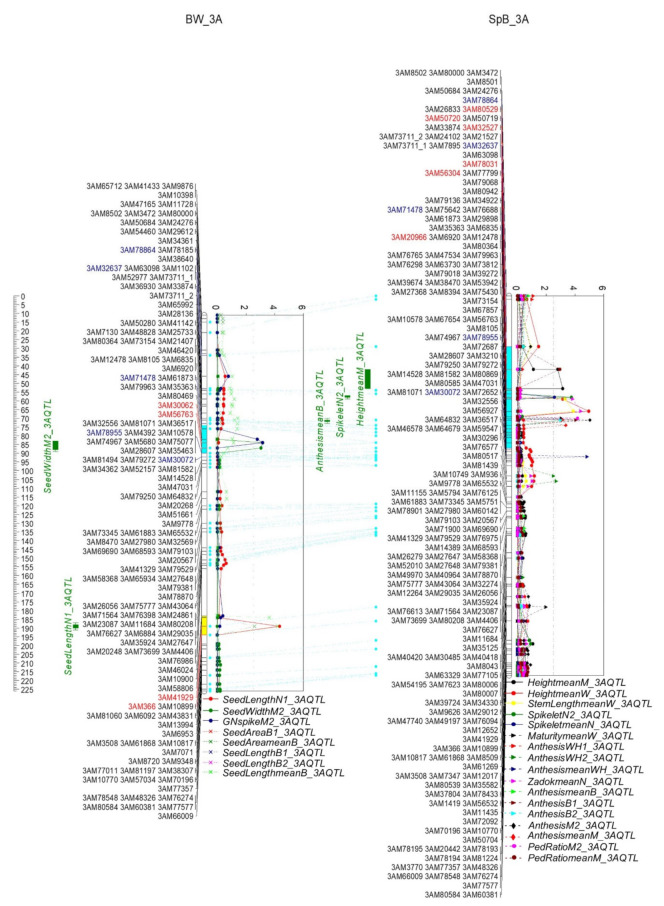
Significant QTL formed two QTL clusters on 3A—one on the homologous region on 3A—in both populations, BW and SpB. Homologous regions in both populations are highlighted in light blue; the QTL cluster on one population is highlighted in yellow; markers for QTL are highlighted in red; homologous markers are highlighted in blue.

**Figure 9 ijms-22-11934-f009:**
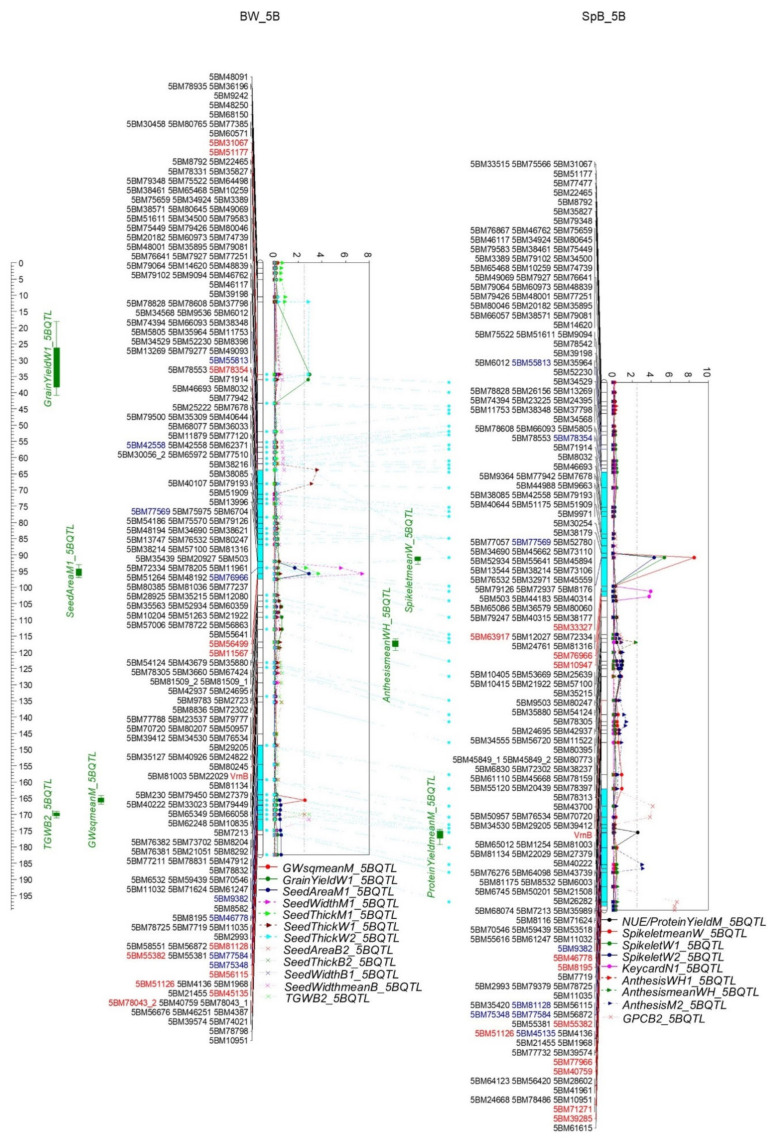
Significant QTL formed two QTL clusters on the homologous regions on 5B in both populations, BW and SpB. Homologous regions in both populations are highlighted in light blue; markers for QTL highlighted in red; homologous markers in blue.

**Figure 10 ijms-22-11934-f010:**
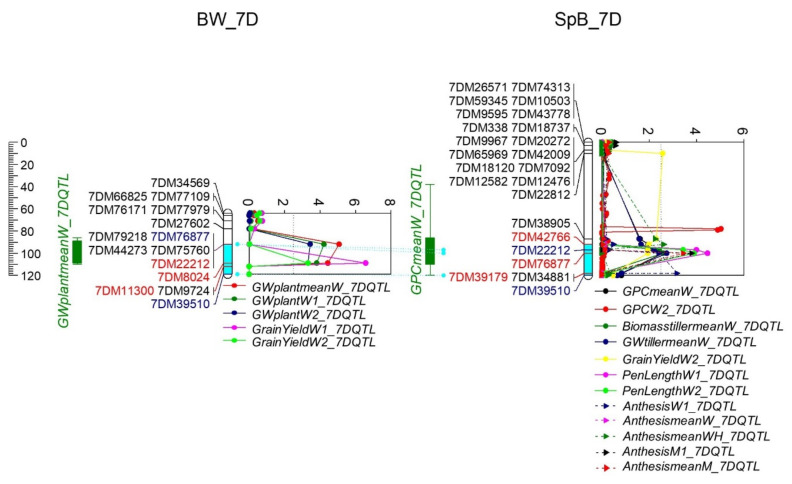
Significant QTL formed one QTL cluster on the homologous region on 7D in both populations, BW and SpB. Homologous regions in both populations are highlighted in light blue; markers for QTL highlighted in red; homologous markers highlighted in blue.

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
