# Peer review of "Yield-Related QTL Clusters and the Potential Candidate Genes in Two Wheat DH Populations"

_ijms, 2021, doi:10.3390/ijms222111934_

Round 1

Reviewer 1 Report

This manuscript by Zhang et al., reports QTLs relating to yield in wheat. The authors carried out QTL analysis by using two DH populations in different environments. They found totally 228 QTLs and found that those QTLs forms clusters.. This study may provide some important information for improving the wheat yield. However, I could not find what is a new finding in this paper, because they said most of all QTLs found in this study have been reported previously (those were summarized in “3.2 Confirmation of QTL clusters and reported genes). I feel that this method itself (for confirmation of QTLs) is strange. In order to confirm the QTLs identified newly in the study, plant geneticists usually select some major QTLs, and carry out linkage analysis (or fine-mapping) of those QTLs. Checking QTLs as the Mendelian factors is confirmation of QTLs. They didn’t do such assay but just checked the correspondence of them based on references. Moreover, those description of the correspondence with the past data should be put in Discussion section but not in Result section (I couldn’t find discussion section in this manuscript). Overall, the manuscript is a wordy explanation of methods, and individual QTLs. For example, the map information should not be described in details. Instead, it would be nice to summarize them in a single Table or Figure, and describe briefly. The authors should make the manuscript more simple description.

Author Response

>>> Thank you for your valuable comments.

The confirmation of QTL clusters in this study validated the current QTL clusters. The QTL cluster validation is particularly important for the candidate gene prediction within the cluster. The new important findings of this MS are 1) identifying the pleiotropic effects on traits of genes within the clusters; 2) pointing out the QTL clusters with consistently positive correlation or without negative impacts on agronomic traits for wheat breeding; 3) figuring out the QTL clusters with consistently negative impacts on agronomic traits which may be valued in special environments. (Text changed in Line 20).

In this journal, the results are the 2nd section (2. Results), and the discussion is the 3rd section, followed by the Materials and Methods (number 4). Therefore, 3.2 Confirmation of QTL clusters and Reported Genes is on the discussion section (number 3).

The map information has been summarized into a table (Table S3), and the text in the map information has been reduced (Line 243 and 248).

Reviewer 2 Report

In this paper, two double haploid populations of bread wheat were phenotyped with 21 traits, including yield components and some grain quality traits. The segregant populations were genotyped to obtain a consensus map. The phenotypic and genotypic data were used to detect QTLs related to yield and grain protein content. In the QTL regions, a search of candidate genes was performed.

The field experiments were carried out in three different locations. In each environment, the all DH lines were not replicated. In addition, the trials were carried out without an appropriate experimental design.

None statistic test was used to compare the phenotypic data between parental lines. This aspect appears necessary because the QTL analysis can be performed only for the traits contrasting between the parental lines.

As reported in the text of results paragraph, the locations and the growing seasons affected the phenotypic data, although none combined analysis was showed in the manuscript.

The maps were obtained on less of 100 DH lines. Currently, the genetic or physical maps of bread wheat are obtained using at least 150 segregant lines.

Why, in the main text are reported the correlation analysis of the trials carried out at Williams and the other correlation are reported as supplementary files? A low quality and unclear format was used in the manuscript and in the supplementary files to display the correlation analysis.

The QTLs analysis must be performed on mean values of location, not can be carried out on the data of each replication.

Usually, the QTL analysis can be performed only using the lines with genotypic and phenotypic data. Considering the low number of DH lines genotyped I think that the QTL analysis was carried out using a very small size of populations.

Finally, several parts of the manuscript resulted unformatted (see reference paragraph).

In conclusion additional genotypic and phenotypic analysis are need. A resulting reorganization of manuscript will be need. Therefore, I suggest a rejection of the manuscript.  

Author Response

In this paper, two double haploid populations of bread wheat were phenotyped with 21 traits, including yield components and some grain quality traits. The segregant populations were genotyped to obtain a consensus map. The phenotypic and genotypic data were used to detect QTLs related to yield and grain protein content. In the QTL regions, a search of candidate genes was performed.

The field experiments were carried out in three different locations. In each environment, the all DH lines were not replicated. In addition, the trials were carried out without an appropriate experimental design.

>>> The field experiments were properly designed based on DiGGer design (Section 4.2, Line 1016 and 1017). The majority of the DH lines (>96%) were replicated at one or more of the three locations (Line 1014 and 1015). The field experimental layout was designed by a professional statistician (Dr. Kefei Chen - a co-author, who is appointed by the Australian GRDC and is responsible for the nationwide field trial designs and data analysis of all GRDC projects) and so does for the data analysis. All field experiments were properly designed according to statistical principles.

None statistic test was used to compare the phenotypic data between parental lines. This aspect appears necessary because the QTL analysis can be performed only for the traits contrasting between the parental lines.

>>>The statistic tests have been done to parental lines both under each environment and across environments. The results have been included in Figure 1 and Table S1, and results were included in the text.

Reviewer’s statement of “This aspect appears necessary because the QTL analysis can be performed only for the traits contrasting between the parental lines” is certainly not correct. It is common knowledge that QTL are detected based segregations. For example, If the two parents of a DH population harbour different Rht genes (one parent with Rht1, another with Rht2), while those heights are similar. The height of the population was segregated, and the QTL were on Rht1 (4B) and Rht2 (4D) (Zhang et al. 2013). Similar results for flowering genes (Vrn1) on 5A, 5B and 5D (Zhang et al., 2021). Parents holding different Vrn1 genes showed one or two day differences, which may not be necessarily statistical different between parents.

As reported in the text of results paragraph, the locations and the growing seasons affected the phenotypic data, although none combined analysis was showed in the manuscript.

>>>Once again all data analysis was carried out by a highly reputed statistician. The QTL analysis have been done in each environment since the differentiations between environments were large. The combination of the data can be done but may not be accurate.

The maps were obtained on less of 100 DH lines. Currently, the genetic or physical maps of bread wheat are obtained using at least 150 segregant lines.

>>> We used two populations that contain 105 and 168 lines (Line 1005). The two populations that has one common parental line. The consensus map was constructed based on all lines in the two populations. It enriched the map and QTL clusters on conserved homoeologous region with candidate gene locations validated, indicating that the current results were robust and reliable.

Why, in the main text are reported the correlation analysis of the trials carried out at Williams and the other correlation are reported as supplementary files? A low quality and unclear format was used in the manuscript and in the supplementary files to display the correlation analysis.

>>>It would be too many if all correlation figures were presented in the main text. The correlation at Williams is the representative correlation results. We enlarged all figures in the revised manuscript.

The QTLs analysis must be performed on mean values of location, not can be carried out on the data of each replication.

>>>The mean values of each location were used in QTL analysis. The label “mean” in Supplementary Table S4 and QTL figures refer to the QTL of mean values.

Usually, the QTL analysis can be performed only using the lines with genotypic and phenotypic data. Considering the low number of DH lines genotyped I think that the QTL analysis was carried out using a very small size of populations.

>>> We used two populations which has one common parental line. The consensus map was constructed by combining the two populations, which are used to mapping the QTL clusters.

Finally, several parts of the manuscript resulted unformatted (see reference paragraph).

 >>>Thanks. We have discussed this with the editorial officers and they have agreed to adjust the reference format.

In conclusion additional genotypic and phenotypic analysis are need. A resulting reorganization of manuscript will be need. Therefore, I suggest a rejection of the manuscript. 

>>> We investigated 22 yield related traits in four environments in two years. As the reviewer 3 pointed out, this is a huge data set (large volume of genotypic and phenotypic data). In each environment, we have two replicates, and all field experiments were based on standard statistical design. We used two populations which held one common parental line. The consensus map enriched the map and QTL clusters on conserved homoeologous region with candidate gene locations validated the robust results.

Reviewer 3 Report

The manuscript entitled with “Yeld related QTL clusters and the potential candidate genes in two wheat DH populations” by Zhang et al. (Manuscript ID: ijms-1399247) described on the identification of potential candidatae genes underly QTL cluster in two DB population. In general, the experiments and manuscript were well conceived and conducted. The results compared with appropriate controls and important finding is presented in results and figures. Appropriate literature cited and discussed all relevant results. The technique is translatable to other systems and this increases the value of the work. Despite the manuscript contained huge data, author’s opinion for QTL results were not enough in Discussion section. Author just enumerated QTL information in the manuscript. There are, however, several points of interest for the readers of this journal. The manuscript handled with large data and provide several valuable information for wheat breeding. Therefore, I recommend this manuscript for the publication in International Journal of Molecular Sciences after incorporating minor revision.

Author Response

>>>Thanks for your value comments. Two more paragraphs have been added to the discussion section.

Line 599-607: Grain yield related agronomic traits have been used as direct selection criteria during wheat breeding based on their high heritability and correlations with grain yield [14]. In the last three decades, molecular markers have been applied as an efficient tool for molecular marker-assisted breeding and effectively shortened the breeding process. Based on the developed molecular markers, QTL mapping of agronomic traits tends to a key approach to identify major QTL and isolate underlying genes in wheat genome [69]. High-density linkage map and wheat genome sequencing provide more precise genetic information for targeted agronomic traits. Consensus map used in this study is one of ways to enrich the genetic information from different populations [11].

Line 617-621: QTL of agronomic traits in different studies are not always the same because of different genetic background of materials, the variations of environments and different cultivation methodologies. Therefore, the QTL validation is crucial through sample replications, different environments, and other QTL studies [3]. The robust QTL or QTL clusters in multiple environments provide solid information for further underlying gene identification.

Reviewer 4 Report

The paper is very important for optimization of wheat marker based selection system, contains interesting results that are well analysed and compared with literature. Groups of genes are revised and number of candidate genes for selection are proposed. The paper is generally well written but I found two main issues that need to be clarified.

Chapter 2.3 contains data on marker density that needs to be revised and corrected. For description of marker density only unique/bin markers should be used, presence of redundant markers only artificially increases density. It is ok, that all segregating markers were kept for consensus map - that is composed of 3322 markers but only 874 are unique/bin markers. These redundant markers are not necessary in figures - but markers connected with QTL were marked with the same colour so figures are informative and no need to correct.

Chapter 2.5 In table S5 'physical positions' of several (31) markers are given in opposite order, and genetic positions of consensus regions are confusing - seems not correct. Let’s take consensus 1A.1 physical region 458397010-494582877 corresponds to 58-78 on BW map and corresponds to different range on consensus map (not 75.1-117.4). Please check it out and correct for another consensus regions. Next consensus region 1A.2 is declared 140.8-end, but please check positions of flanking markers on consensus map (Table S2), all map to position of 140.8! Therefore positions of markers listed as flanking QTLs should be checked and corrected prior to publishing.

Some minor editorial errors also occurs:
unjustified italics lines 75-77, 638-640
spelling mistakes i.e.: lines 121, 257, 583, 615
Chapter of references contain no spaces.

Figure 2 and S2. Abbreviations of genotypes are provided but not present in these figures.

Author Response

The paper is very important for optimization of wheat marker based selection system, contains interesting results that are well analysed and compared with literature. Groups of genes are revised and number of candidate genes for selection are proposed. The paper is generally well written but I found two main issues that need to be clarified.

Chapter 2.3 contains data on marker density that needs to be revised and corrected. For description of marker density only unique/bin markers should be used, presence of redundant markers only artificially increases density. It is ok, that all segregating markers were kept for consensus map - that is composed of 3322 markers but only 874 are unique/bin markers. These redundant markers are not necessary in figures - but markers connected with QTL were marked with the same colour so figures are informative and no need to correct.

>>>Thanks for your comments. We have revised the manuscript based on the suggestions. Bin markers density have been used. A table of “The total marker number, bin marker density and map coverage on the maps of BW, SpB and consensus map BW-SpB” were added to Supplementary Table S3. The text has been changed based on bin marker density.

Chapter 2.5 In table S5 'physical positions' of several (31) markers are given in opposite order, and genetic positions of consensus regions are confusing - seems not correct. Let’s take consensus 1A.1 physical region 458397010-494582877 corresponds to 58-78 on BW map and corresponds to different range on consensus map (not 75.1-117.4). Please check it out and correct for another consensus regions. Next consensus region 1A.2 is declared 140.8-end, but please check positions of flanking markers on consensus map (Table S2), all map to position of 140.8! Therefore positions of markers listed as flanking QTLs should be checked and corrected prior to publishing.

>>>Thanks for pointing out the error. Table S5 (now Table S6) has been checked and corrected. The main text has been changed accordingly. The physical map range refers to the minimum and maximum physical map location of SNP markers within the cluster in IWGSC RefSeq v1.0. SNP markers in the clusters were used based on the SNP marker order in the consensus map. Two more footnotes were added in Table S5 (now Table S6). The text has been corrected thoroughly.

Some minor editorial errors also occurs:
unjustified italics lines 75-77, 638-640
spelling mistakes i.e.: lines 121, 257, 583, 615
Chapter of references contain no spaces.

>>>Thanks, most errors pointed out have been fixed. Could the reviewer please point out the details of spelling mistakes on lines 257 and 615.

We have discussed the references with the editorial officers and they have agreed to adjust the reference format.

Figure 2 and S2. Abbreviations of genotypes are provided but not present in these figures.

>>>Thanks, the mistakes have been corrected.

Round 2

Reviewer 1 Report

Now it's all right.